# In situ structure of actin remodeling during glucose-stimulated insulin secretion using cryo-electron tomography

Weimin Li[1,2], Angdi Li[1,2], Bing Yu[1,2], Xiaoxiao Zhang[1], Xiaoyan Liu[1], Kate L. White[3], Raymond C. Stevens[1,2], Wolfgang Baumeister [1,4] ✉, Andrej Sali [5,6,7] ✉, Marion Jasnin [8,9] ✉ & Liping Sun [1] ✉

Actin mediates insulin secretion in pancreatic β-cells through remodeling. Hampered by limited resolution, previous studies have offered an ambiguous depiction as depolymerization and repolymerization. We report the in situ structure of actin remodeling in INS-1E β-cells during glucose-stimulated insulin secretion at nanoscale resolution. After remodeling, the actin filament network at the cell periphery exhibits three marked differences: 12% of actin filaments reorient quasi-orthogonally to the ventral membrane; the filament network mainly remains as cell-stabilizing bundles but partially reconfigures into a less compact arrangement; actin filaments anchored to the ventral membrane reorganize from a "netlike" to a "blooming" architecture. Furthermore, the density of actin filaments and microtubules around insulin secretory granules decreases, while actin filaments and microtubules become more densely packed. The actin filament network after remodeling potentially precedes the transport and release of insulin secretory granules. These findings advance our understanding of actin remodeling and its role in glucose-stimulated insulin secretion.

Actin filaments are essential for a variety of cellular functions, including maintaining the cell structure, driving cell migration and division, and governing the movement of subcellular components, such as mitochondria, lysosomes, and insulin secretory granules (ISGs)[1,2]. Cellular signaling pathways, such as Rho GTPase pathway and PI3K/Akt pathways, cause the remodeling of actin filaments[3,4]. An early 2D visualization of actin remodeling, starting with a parallel alignment and ending in an "unidentifiable dense mass", was achieved in vitro for chemically fixed β-cells using conventional electron microscopy

(EM) in 1972[5]. Since then, actin remodeling has been studied at an even lower resolution by fluorescence imaging and by western blot analysis, revealing that actin filaments first depolymerize into actin monomers (G-actin) and then repolymerize into actin filaments (F-actin)[5–10]. To date, the in situ structure of the actin filaments before and after remodeling has yet to be established at the nanoscale.

Glucose-stimulated insulin secretion (GSIS) in pancreatic β-cells is a biphasic process defined by the presence of two distinct pools of ISGs: readily releasable pool (RRP) and reserve pool (RP)[7]. ISGs in RRP

[1]iHuman Institute, ShanghaiTech University, Shanghai 201210, China. [2]School of Life Science and Technology, ShanghaiTech University, Shanghai 201210, China. [3]Department of Chemistry, Bridge Institute, USC Michelson Center for Convergent Bioscience, University of Southern California, Los Angeles, CA 90089, USA. [4]Department of Molecular Structural Biology, Max Planck Institute of Biochemistry, 82152 Martinsried, Germany. [5]Quantitative Biosciences Institute, University of California, San Francisco, San Francisco, CA 94158, USA. [6]Department of Bioengineering and Therapeutic Sciences, University of California, San Francisco, San Francisco, CA 94158, USA. [7]Department of Pharmaceutical Chemistry, University of California, San Francisco, San Francisco, CA 94158, USA. [8]Helmholtz Pioneer Campus, Helmholtz Zentrum München, 85764 Neuherberg, Germany. [9]Department of Chemistry, Technical University of Munich, 85748 Garching, Germany. ✉e-mail: baumeist@biochem.mpg.de; sali@salilab.org; marion.jasnin@helmholtz-munich.de; sunlp@shanghaitech.edu.cn

are rapidly released within the first 5–10 minutes, corresponding to the first phase, followed by a prolonged second phase lasting hours[11,12]. Actin filaments are recognized as a key mediator of GSIS through observation of the restricted ISG movements during actin stabilization by drugs via fluorescence microscopy[8,13]. Through remodeling, actin filaments regulate the transport of ISGs from the cell periphery to the plasma membrane (PM) and the release of their contents into the extracellular space[8–10,14]. Under basal conditions, actin filaments act as a barrier parallel to the PM to block ISGs from approaching the PM[8,10]. In the first phase, actin filaments depolymerize into actin monomers, allowing the rapid release of ISGs in RRP[8,9]. In the second phase, actin filaments repolymerize into a new actin network that facilitates ISGs to transport to and release from the PM[8,9,13]. Quantifying the differences between the actin filament network before and after remodeling is imperative to understand the mechanisms underlying its blocking or facilitating role during GSIS.

During GSIS, the transportation of ISGs first involves long-range movement driven by kinesins on microtubules (MTs), with the subsequent handoff to myosin V for the short-range movement on actin filaments to the PM[15,16]. Using fluorescence microscopy, MTs have been reported to negatively regulate insulin secretion at the cell periphery under basal conditions[17]; it was proposed that direct interaction between actin filaments and MTs promotes the transport and release of ISGs in the second phase[10]. Due to the limited resolution of fluorescence microscopy, the structural basis of interactions among ISGs, actin filaments, and MTs during GSIS remains to be determined.

In this work, we used a multimodal imaging approach to visualize and quantify actin remodeling at both the periphery and interior of the cell under basal conditions as well as during the first and second phases of insulin secretion in INS-1E β-cells. First, we assessed biphasic insulin secretion using both an enzyme-linked immunosorbent assay (ELISA) experiment and total internal reflection fluorescence (TIRF) imaging. We quantified changes in actin meshwork organization during GSIS using structured illumination microscopy (SIM). Next, we employed cryo-electron tomography (cryo-ET) to reveal the structure of actin remodeling at the nanoscale and variations in the volume and orientation of actin filaments in both INS-1E β-cells and rat primary β-cells during GSIS[18–20]. We then conducted SIM imaging of INS-1E β-cells treated with Y15 to elucidate the underlying mechanisms and functional significance of actin angle changes during remodeling. In addition, we quantified the arrangement of peripheral actin filaments in INS-1E β-cells before and after remodeling by their distances. We analyzed the alignment of peripheral actin filaments anchored to the ventral membrane (VM) via their distances and angles. Lastly, we characterized the interactions among ISGs, actin filaments, and MTs at the periphery of INS-1E β-cells by their distances. Based on these quantitative analyses, we established a model for actin remodeling at the cell periphery, providing a more detailed and accurate depiction of changes in the architecture, alignment, and interaction of actin filaments with ISGs and MTs during GSIS in pancreatic β-cells.

## Results

### Multimodal visualization of actin remodeling during glucose-stimulated insulin secretion

We imaged clonal INS-1E β-cells, a rat β-cell widely used to mimic the biphasic insulin secretion in β-cells in response to glucose stimulation[21–23]. We began by multimodal imaging to visualize actin remodeling during GSIS. We first performed an ELISA experiment to detect the level of insulin secretion in both INS-1E β-cells and rat primary β-cells under different conditions. We then employed TIRF microscopy (x and y resolution of ~120 nm) to capture the biphasic insulin secretion response over 40 minutes of glucose stimulation in both INS-1E β-cells and rat primary β-cells (Supplementary Fig. 2 and Supplementary Movies 1 and 2). The first phase occurred during the first 5 minutes after glucose stimulation, while the second phase

occurred 30 minutes after glucose stimulation, as we recorded the intensity of the NPY-mCherry marker for ISGs over time. Actin stress fibers were observed at the VM, maintaining the cell structure. We then performed a western blot experiment to detect the G-/F-actin ratio in the cell. We observed a significant increase in the ratio of the monomeric to filamentous actin states under the 16.7 Glu – 5 min condition, followed by a significant decrease under the 16.7 Glu – 30 min condition, indicating the remodeling of actin filaments during each phase (Supplementary Fig. 3).

Next, we performed SIM (x and y resolution of ~30 nm) to assess the actin organization around ISGs in different subsections of the β-cell. Note that we characterize an actin meshwork, as individual filaments are not distinguishable in SIM due to its limited resolution. We visualized the projections of actin and ISGs along the z-axis of a 300 nm-thick cell section near the VM (Fig. 1a–c). We observed a number of bundles under basal conditions (44% ± 6%, bundle area by total actin area) and the 16.7 Glu – 30 min condition (30% ± 3%), but few under the 16.7 Glu – 5 min condition (18% ± 1%). Compared to the basal conditions, actin intensity significantly decreases under the 16.7 Glu – 5 min condition, followed by an increase under the 16.7 Glu – 30 min condition (Fig. 1a–f). These changes are in agreement with changes in the G-/F-actin ratio measured by western blot (Supplementary Fig. 3). In addition, they are consistent with similar measurements by western blot and immunofluorescence techniques in previous studies[24,25]. To quantify the structural properties of actin meshwork organization during GSIS, we selected a few subsections of actin surrounding ISGs rather than other areas where actin forms bundles in the cell (Supplementary Fig. 4). After skeletonizing the actin meshwork based on the intensity of the actin fluorescence label[26–28] ("Methods", Fig. 1g), we quantified the length of the actin meshwork and identified network junctions as connections in the actin meshwork. The actin meshwork is significantly shorter under the 16.7 Glu – 5 min condition (515 ± 53 nm) and significantly longer under the 16.7 Glu – 30 min condition (2435 ± 167 nm) compared to the basal conditions (1432 ± 116 nm), in line with the western blot measurements (Fig. 1h and Supplementary Fig. 3). Notably, actin forms a more complex network with an increased number of junctions under the 16.7 Glu – 30 min condition (9.4 ± 0.7) compared to the basal conditions (6.5 ± 0.4) and the 16.7 Glu – 5 min condition (4.7 ± 0.5) (Fig. 1i). In summary, these results characterize actin remodeling near the VM during GSIS, including depolymerization under the 16.7 Glu – 5 min condition as well as repolymerization into a longer and more complex actin meshwork under the 16.7 Glu – 30 min condition.

Next, we used cryo-ET to examine actin remodeling during GSIS at nanometer resolution, and quantify changes in architecture, alignment, and interaction of actin filaments with other subcellular components. We collected tomograms of the cell periphery (0–6 μm from the PM) and the cell interior (0–3 μm from the nuclear membrane) (Supplementary Table 2). The cell periphery of INS-1E β-cells is only ~250 nm thick, and vitrified cells can be imaged directly using cryo-ET (Supplementary Fig. 6). To access the cell interior, vitrified INS-1E β-cells were first milled using a cryo-focused ion beam (cryo-FIB) machine to obtain lamellas with a thickness of ~150 ± 50 nm suitable for cryo-ET data collection[29,30] (Supplementary Fig. 7). In total, we collected 42 tomograms in 34 INS-1E β-cells, including 24 tomograms at the cell periphery and 18 tomograms at the cell interior under basal conditions (2.8 Glu – 30 min), 16.7 Glu – 5 min and 16.7 Glu – 30 min conditions; 8 tomograms in 6 rat primary β-cells, all at the cell periphery under basal conditions and the 16.7 Glu – 30 min condition (Supplementary Table 2).

In tomograms of the cell periphery, we distinguished various types of subcellular components, including single actin filaments[31,32]; MTs as rigid tubular structures[33,34]; mature and immature ISGs as membrane enclosed vesicles with and without dense core,

respectively[35]; lysosomes as monolayer vesicles with several types of contents in the lumen[36]; and ribosomes (Fig. 2a–c). We labeled these components via automated segmentation followed by manual refinement (Fig. 2d–f and Supplementary Movies 3–5). Specifically, we applied a cylinder cross-correlation algorithm to segment actin filaments and MTs[37], tomosegmemtv to segment membranous organelles such as ISGs and lysosomes[38], and template matching to map ribosomes[39]. We define the segmentation unit as 60 nm for actin filaments and 100 nm for MTs.

We observed changes in the amounts of actin filaments and MTs in the tomograms of the cell periphery under different conditions (Fig. 2a–f), in accordance with previous fluorescence studies[25,40]. Specifically, both the volume ratio of actin filaments to the tomogram and the volume ratio of MTs to the tomogram significantly decrease under the 16.7 Glu – 5 min condition, indicating a nearly complete depolymerization. The volume ratio of actin filaments at the cell periphery after repolymerization does not change significantly compared to the basal conditions, in contrast to observations from randomly selected

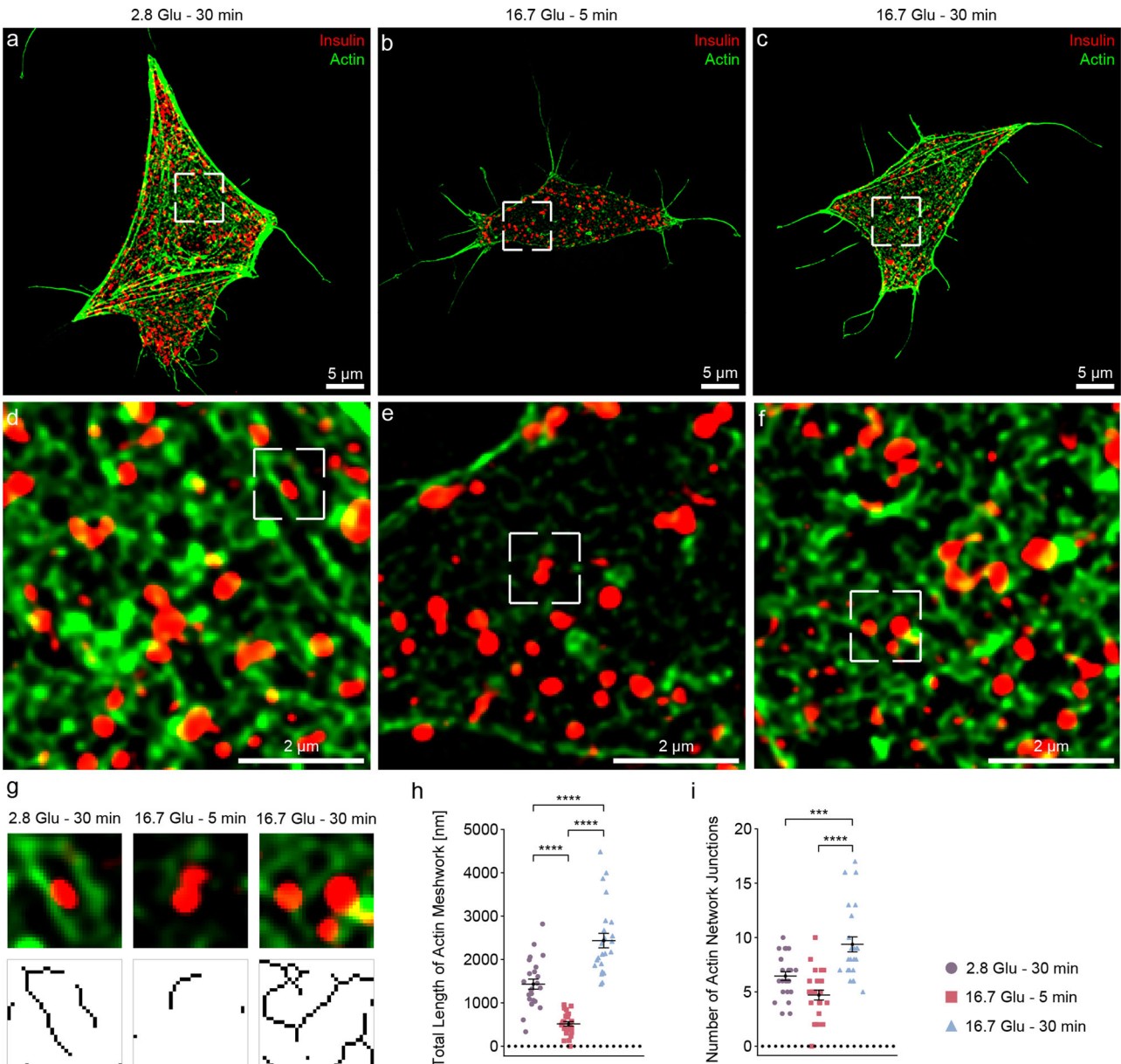

**Fig. 1 | Visualization and quantification of the actin meshwork organization near the VM during GSIS using SIM.** SIM images of actin (green) and ISGs (red) in INS-1E β-cells under basal conditions (**a**, **d**) and after 5 minutes (**b**, **e**) or 30 minutes (**c**, **f**) of 16.7 mM glucose stimulation. **a**–**c** 2D projection of a 300 nm-thick volume from the VM. **d**–**f** Zoomed-in views showing actin meshworks peripheral to ISGs. The images for subsequent skeletonization analysis are also highlighted with a small white rectangle of 0.97 um². **g** Actin and ISG images on selected subsections and corresponding skeletonized actin meshworks. For further analysis, only actin bundles with lengths longer than or equal to one pixel were considered. **h** Total length of actin meshwork in each subsection under different conditions. 2.8

Glu – 30 min versus 16.7 Glu – 5 min, ****$p < 0.0001$; 2.8 Glu – 30 min versus 16.7 Glu – 30 min, ****$p < 0.0001$; 16.7 Glu – 5 min versus 16.7 Glu – 30 min, ****$p < 0.0001$. **i** Number of actin network junctions in each subsection under different conditions. 2.8 Glu – 30 min versus 16.7 Glu – 30 min, ***$p = 0.0007$; 16.7 Glu – 5 min versus 16.7 Glu – 30 min, ****$p < 0.0001$. * indicates $p < 0.05$. ** indicates $p < 0.01$. *** indicates $p < 0.001$. **** indicates $p < 0.0001$ by one-way ANOVA. $n = 24$ subsections, corresponding to four subsections per cell, with six cells used in each condition. For each condition, a total of six SIM images were collected from individual INS-1E β-cells from three biologically independent experiments. Data are presented as mean values ± SEM. Source data are provided as a Source Data file.

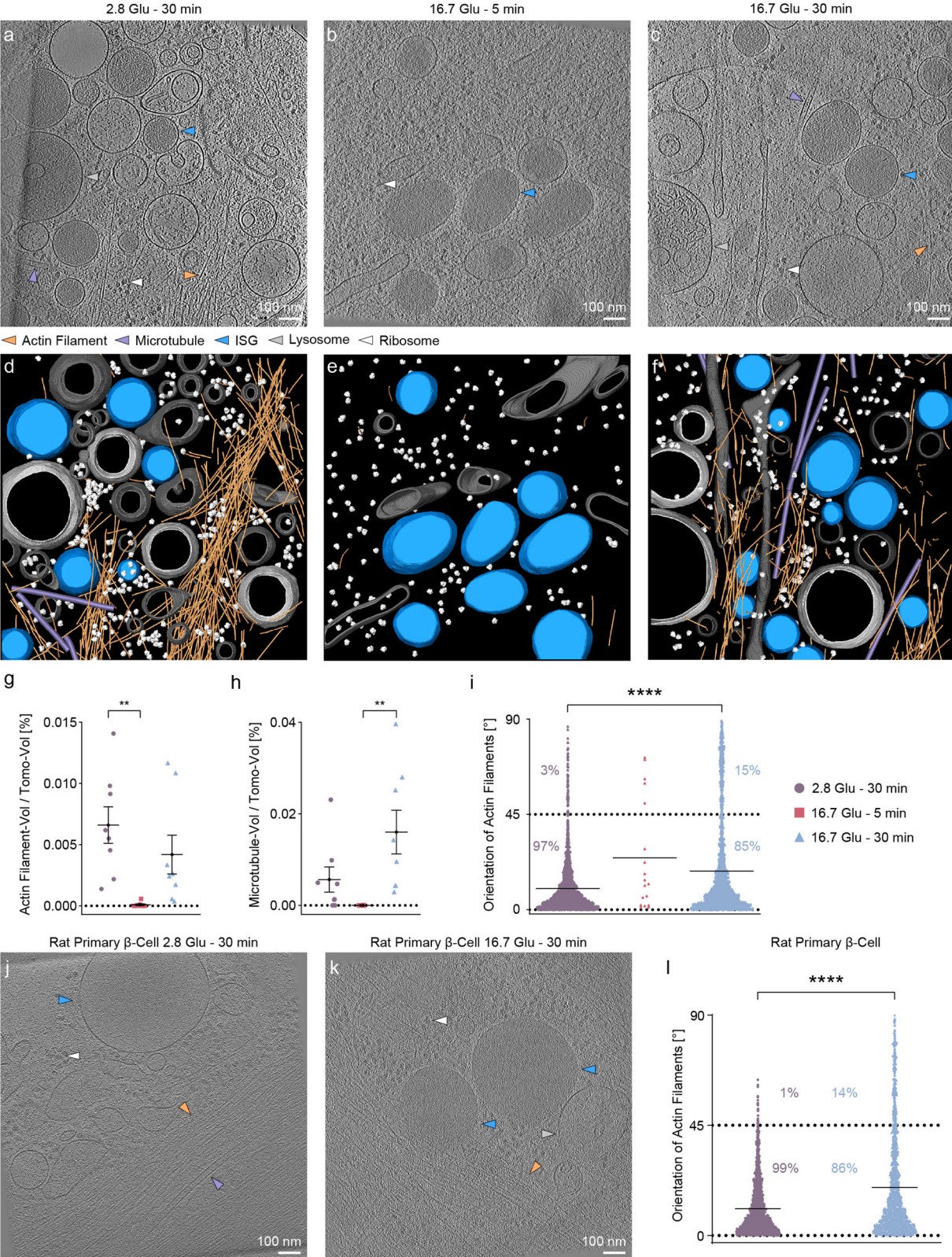

subsections of the cell in fluorescence images acquired above (Figs. 2g, h and 1h). This analysis provides a direct in situ quantification of the remodeling of actin filaments and MTs at the cell periphery.

Actin filaments maintain cell structure by forming bundles parallel to the VM at the cell periphery[41]. They also regulate the transport and release of ISGs through a remodeling process, as proposed in previous studies using fluorescence microscopy[8,13].

We computed changes in the orientation of individual actin filaments relative to the VM (which lies in the XY plane of a tomogram) (Fig. 2 and Supplementary Fig. 8). Under basal conditions, 97% of the actin filaments are oriented at angles of 0–45° to the VM (i.e., semi-parallel orientation). Under the 16.7 Glu – 30 min condition, the amount of actin filaments oriented at angles of 45–90° to the VM (i.e., quasi-orthogonal orientation) increases from 3% to 15% (Fig. 2i);

**Fig. 2 | Quantitative analysis of actin filament and MT orientations relative to the VM at the cell periphery during GSIS. a–c** Slices through INS-1E β-cell tomograms under basal conditions (2.8 Glu – 30 min (**a**), the 16.7 Glu – 5 min condition (**b**), and the 16.7 Glu – 30 min condition (**c**). Arrows indicate actin filaments (orange), MTs (violet), ISGs (blue), lysosomes (silver), and ribosomes (white). **d–f** Respective segmentations with the same color code. Non-ISG membranes are shown in gray. **g, h** Volume ratio of actin filaments (**g**) or MTs (**h**) to the tomogram in each INS-1E β-cell tomogram under different conditions. 2.8 Glu – 30 min versus 16.7 Glu – 5 min, **p = 0.04 in (**g**). 16.7 Glu – 5 min versus 16.7 Glu – 30 min, **p = 0.0051 in (**h**). **i** Orientation of actin filaments relative to the VM under different conditions in INS-1E β-cell tomograms. 2.8 Glu – 30 min versus 16.7 Glu – 30 min, ****p < 0.0001. **j, k** Slices through rat primary β-cell tomograms under basal conditions (**j**), and the 16.7 Glu – 30 min condition (**k**). **l** Orientation of actin filaments relative to the VM under different conditions in rat primary β-cell tomograms. 2.8 Glu – 30 min versus 16.7 Glu – 30 min, ****$p < 0.0001$. * indicates $p < 0.05$. ** indicates $p < 0.01$. *** indicates $p < 0.001$. **** indicates $p < 0.0001$ by one-way ANOVA. $n = 24$ tomograms, corresponding to eight tomograms for each condition from three biologically independent experiments. $n = 8$ tomograms in rat primary β-cells, corresponding to four tomograms for each condition from three biologically independent experiments. The tilt axis of the transmission electron microscope (TEM) corresponds to the Y-axis in the tomograms under basal condition and the 16.7 Glu – 30 min condition (Supplementary Table 3). As shown in Supplementary Fig. 8, actin filament orientations are distributed over the full angular range, including at orientations close to the XZ plane, highlighting the differences between the patterns of the two experimental conditions. Data are presented as mean values ± SEM. Source data are provided as a Source Data file.

the amount of actin filaments with angles of 60–90° to the VM increases from 1% to 11%.

In addition, after confirming that rat primary β-cells exhibit glucose-stimulated biphasic insulin secretion (Supplementary Figs. 1 and 2 and Supplementary Movie 2), we performed cryo-ET on those cells under two conditions: 2.8 Glu – 30 min and 16.7 Glu – 30 min. In tomograms of rat primary β-cells, comparable changes in the orientation of actin filaments relative to the VM are seen during GSIS (Fig. 2j, k and Supplementary Fig. 6, "Methods"). Specifically, we observe a decrease from 99% to 86% in the proportion of actin filaments at angles of 0–45°, and an increase from 1% to 14% in the proportion of actin filaments at angles of 45–90° (Fig. 2l). Together with previous physiological studies[21], this structural similarity further validates the ability of INS-1E β-cells to replicate characteristics of rat primary β-cells. Given the limited number of rat primary β-cell tomograms ($n = 4$ per condition) as compared to INS-1E β-cell tomograms ($n = 8$ per condition), our subsequent quantitative analysis is exclusively carried out using INS-1E β-cell tomograms to ensure the statistical robustness of the results.

In tomograms of the cell interior, we distinguished different types of subcellular components, including actin filaments, MTs, endoplasmic reticulum (ER) as a continuous monolayer membrane[42,43], mitochondria as double-membrane structures with a flat outer membrane and a curved interior membrane[35,44], nuclear membrane[45] and ribosomes[46] (Fig. 3a–c). We labeled these components using the same methods as for the tomograms of the cell periphery (Fig. 3d–f and Supplementary Movies 6–8). In the cell interior, we observed no substantial difference in either the amounts of actin filaments and MTs or the orientations of actin filaments relative to the VM under different conditions (Fig. 3g–i). Thus, we did not observe actin remodeling in the cell interior. The number of actin filaments is insufficient to demonstrate statistically significant differences between the different conditions. Similarly, there are no statistically significant differences in the distances between the cellular components studied here.

In summary, peripheral actin filaments undergo a nearly complete depolymerization under the 16.7 Glu – 5 min condition and then a repolymerization during GSIS. More importantly, more actin filaments are oriented quasi-orthogonally to the VM after remodeling than before remodeling (3% and 15% for the basal conditions and the 16.7 Glu – 30 min condition, respectively). In contrast, we do not observe actin remodeling in the cell interior. Therefore, we proceeded with the analysis of the cell periphery tomograms under basal conditions and the 16.7 Glu – 30 min condition. The analysis of the cell periphery tomograms under the 16.7 Glu – 5 min condition is exclusively presented in the Supplementary Information due to its statistical insufficiency, arising from the limited amount of segmented actin filaments (Supplementary Fig. 9).

## The architecture of the actin filament network before and after remodeling

Tomograms of the cell periphery show a change in the architecture of the actin filament network during remodeling, despite a similar volume of actin filaments (Figs. 2g and 4a–d). We computed the Actin-Actin distance as a function of the actin filament angles ("Methods"). Most actin filaments form bundles, presumably to fulfill their primary function of maintaining cell structure, with an angle of 0–15° and a distance of 12–13 nm, as found in filopodia and stress fibers in epithelial cells[32] (Fig. 4e, f). However, we observe fewer bundles under the 16.7 Glu – 30 min condition compared to basal conditions (yellow-to-red circled region in Fig. 4e, f). In addition, actin filaments oriented parallel to the VM maintain the same distance distribution after remodeling, whereas actin filaments oriented quasi-orthogonally to the VM show a shift to longer distances under the 16.7 Glu – 30 min condition compared to the basal conditions (Fig. 4g, h). These results indicate that actin filaments mostly retain their bundled formation but partially reconfigure into a less compact arrangement that may be required to transport subcellular components from the cell periphery to the VM.

Focal adhesion complexes are known to mediate interactions between actin filaments and the PM[8,47,48]. We performed SIM imaging of INS-1E β-cells treated with Y15, a focal adhesion kinase inhibitor, to explore the molecular mechanisms driving actin angle alterations during remodeling (Fig. 4i and Supplementary Table 1). We selected the actin meshwork located within 300 nm of the VM and computed their angles relative to the VM (Methods). After remodeling, we observe a significant decrease in the amount of actin filaments oriented at angles of 0–15° relative to the VM, and an increase in the amount of actin with angles of >45° both in cryo-electron tomograms (Fig. 4j) and in SIM images (Fig. 4k). Notably, Y15 treatment abolishes changes in actin angles between before and after remodeling as observed with SIM (Fig. 4l). Given Y15's established ability to reduce insulin secretion in β-cells by 76.9 ± 5.4%[49], we propose that the actin angle alterations during remodeling are facilitated by focal adhesion complexes, and are of crucial functional significance for insulin secretion in β-cells. Disruption of this process impairs actin remodeling during GSIS, subsequently resulting in decreased insulin secretion.

Next, we manually labeled the VM and considered actin filaments within 60 nm of the VM as being anchored to the VM (Fig. 5a–d). The distance is defined as the shortest distance between the end points of each actin filament and the VM; this distance threshold takes into account the length of the actin filament segmentation unit (60 nm). Under basal conditions and in the 16.7 Glu – 30 min condition, the height of the cell periphery (190.5 ± 16.9 nm and 258.4 ± 31.3 nm) remained constant, as well as the number and percentage of anchored actin filaments (263 ± 55 and 183 ± 68; 90% ± 4% and 71% ± 8%), as determined by a t-test (Supplementary Fig. 10). We calculated the angles between neighboring actin filament vectors that were anchored to the VM and mapped these angles against the orientations of the anchored actin filaments relative to the VM (Fig. 5e–h, "Methods"). The filament vector is defined as extending from the end point near the VM to the end point further away from it; neighboring filament vectors are defined as those anchored to the same VM whose end points near the VM are within

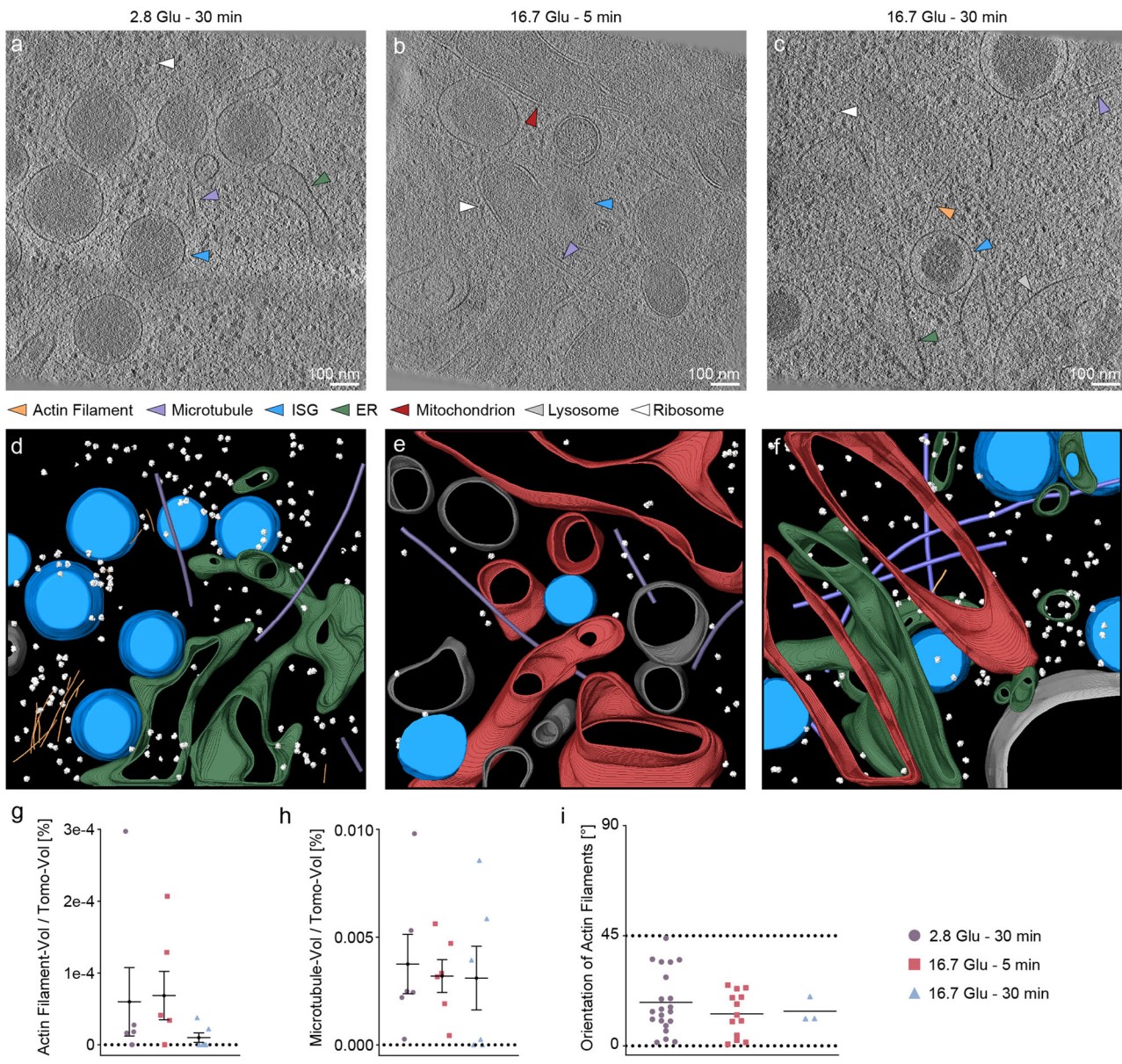

**Fig. 3 | Quantitative analysis of actin filament and MT orientations relative to the VM in the cell interior during GSIS. a–c** Reconstructed tomograms under basal conditions (2.8 Glu – 30 min (**a**), the 16.7 Glu – 5 min condition (**b**), and the 16.7 Glu – 30 min condition (**c**). Arrows indicate actin filaments (orange), MTs (violet), ISGs (blue), ER (green), mitochondria (red), lysosomes (silver), and ribosomes (white). **d–f** Respective segmentations with the same color code. Non-ISG membranes are shown in gray. **g, h** Fraction of actin filaments (**g**) or MTs (**h**) in each tomogram under different conditions. **i** Orientation of actin filaments relative to the VM under different conditions. * indicates $p < 0.05$. ** indicates $p < 0.01$. *** indicates $p < 0.001$. **** indicates $p < 0.0001$ by one-way ANOVA. $n = 18$ tomograms, corresponding to six tomograms for each condition from three biologically independent experiments. The tilt axis of the TEM corresponds to the X-axis in the tomograms. Data are presented as mean values ± SEM. Source data are provided as a Source Data file.

twice the length of the actin filament segmentation unit. Anchored actin filaments are primarily oriented parallel to the VM at angles of 0–15°, with the highest and next highest population of filament alignment at angles of 0–30° and 150–180° (Fig. 5e, f). This indicates a parallel alignment of the anchored actin filaments both with the VM and with each other (i.e., bundles). Under basal conditions, actin filaments oriented parallel to the VM (0–15° population) have a higher population of actin filament angles at 60–120° than the 16.7 Glu – 30 min condition, suggesting a "netlike" architecture, which may act as a barrier preventing ISGs from approaching the VM (Fig. 5e, f, black circle and Fig. 5g, h). Actin filaments oriented quasi-orthogonally to the VM (45–90°) show a clear shift to smaller actin filament angles (0–30°) under the 16.7 Glu – 30 min condition

compared with basal conditions (Fig. 5e, f dashed circle and Fig. 5g, h). These findings suggest that actin filaments reorient quasi-orthogonally to the VM and parallel to each other after remodeling compared to basal conditions. This reorganization changes the actin filament network from a "netlike" to a "blooming" architecture, where radial projections emanate from anchor points at the VM (Fig. 5b, d). We hypothesize that the "blooming" architecture might act as transportation tracks to the VM, which is expected to facilitate the transport and release of ISGs at the VM. These results offer the first structural evidence for the previous hypothesis of actin filaments blocking and facilitating ISG release at the VM before and after remodeling, respectively[5,8,10].

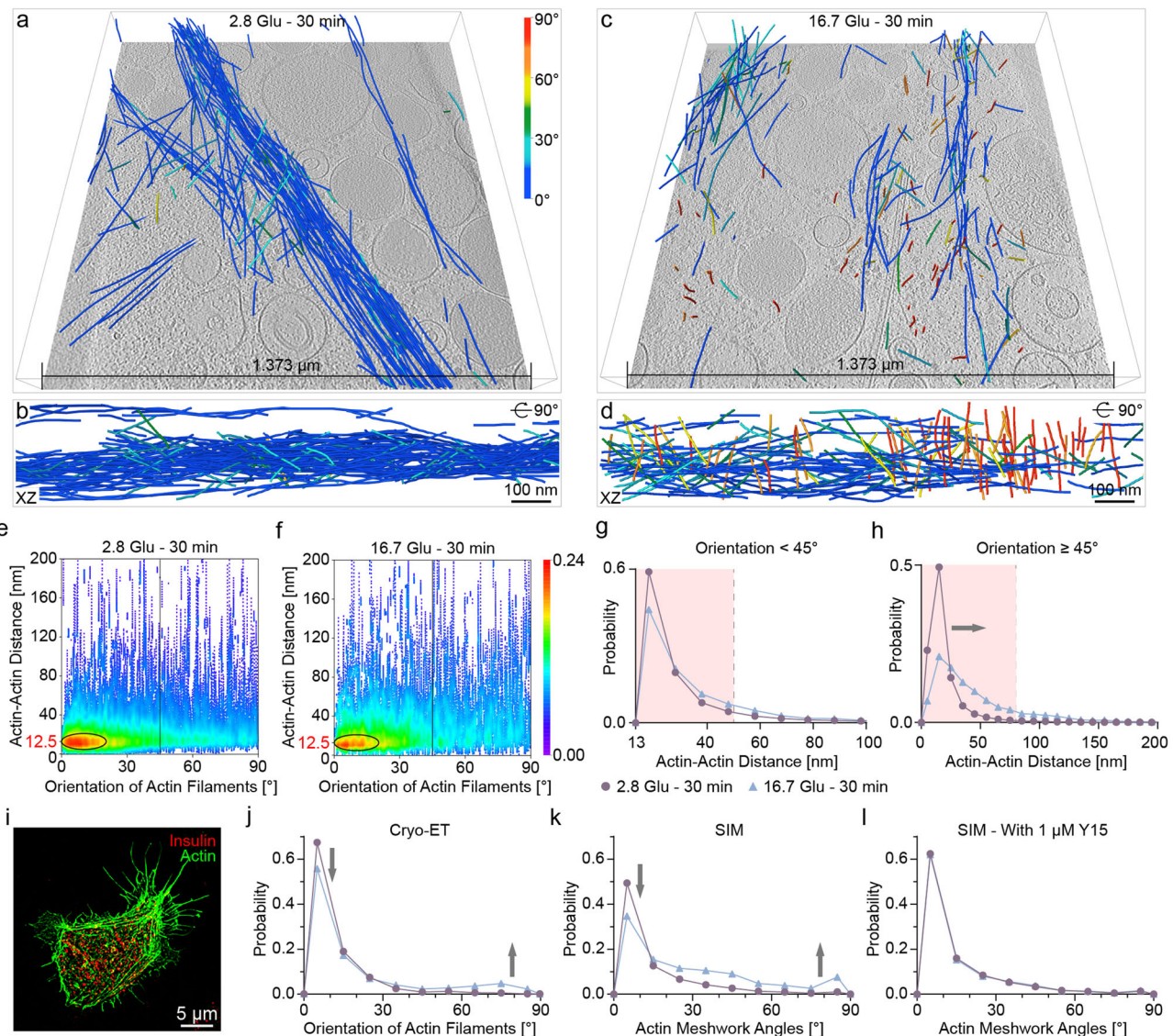

**Fig. 4 | Quantitative analysis of actin filament organization at the cell periphery. a–d** Actin filaments represented as a function of their orientation relative to the VM in a blue-to-red color map (shown in **a**), before (**a**, **b**) and after (**c**, **d**) remodeling in two different views. **e, f** Density map of the distance between actin filaments as a function of the angle between actin filaments before (**e**) and after (**f**) remodeling. **g** Histogram of the distances between actin filaments whose orientation relative to the VM is less than 45° before and after remodeling. Bin size = 10. **h** Histogram of the distances between actin filaments whose orientation relative to the VM is greater than or equal to 45° before and after remodeling. Bin size = 10.

**i** An example SIM image of actin (green) and ISGs (red) in an INS-1E β-cell under the 2.8 Glu – 30 min condition, treated with 1 μM Y15. **j–l** Histograms of the actin angles relative to the VM before and after remodeling, obtained by cryo-ET (**j**), SIM without Y15 treatment (**k**), and SIM with 1 μM Y15 treatment (**l**). Bin size = 10. For each condition, a total of ten SIM images were collected from individual INS-1E β-cells from three biologically independent experiments. Each point on the density map reflects the corresponding density values by calculating kernel density. Source data are provided as a Source Data file.

## Interaction of ISGs, actin filaments, and MTs at the cell periphery

Previous studies demonstrated that actin filaments interact with MTs to facilitate insulin secretion[10]. Specifically, the transport of ISGs occurs along MTs from the cell interior to the cell periphery, then along actin filaments from the cell periphery to the VM[10]. In tomograms of the cell periphery, we observed a few ISGs in vicinity of both actin filaments (Fig. 6a–d) and MTs (Fig. 6e–h). In addition, we observed several ISGs particularly close to the reoriented actin filaments under the 16.7 Glu – 30 min condition, suggesting a potential association with the transport process (Fig. 6b, d). These observations led us to quantitatively analyze the shortest distances between ISGs and actin filaments as well as between ISGs and MTs, as follows.

To characterize interactions between ISGs and actin filaments, we focused on variations in their shortest distances within 200 nm during GSIS. The shortest distance is calculated as the distance between the resampled points of actin filaments (at 4 nm intervals) and the nearest ISG surface ("Methods"). This distance threshold is estimated by considering the length of the actin filament segmentation unit (60 nm) and the lengths of myosin V and other unknown associated proteins (~55 nm for the structurally defined part of myosin V[16] and an additional tolerance of 85 nm to account for the uncertainty resulting from the unknown structure of 268 residues of myosin V and unknown associated proteins). The probability for the shortest distance to be within 200 nm decreases under the 16.7 Glu – 30 min condition compared with basal conditions (Fig. 6i). We mapped the shortest distances along different orientations of actin filaments relative to the VM. Actin filaments

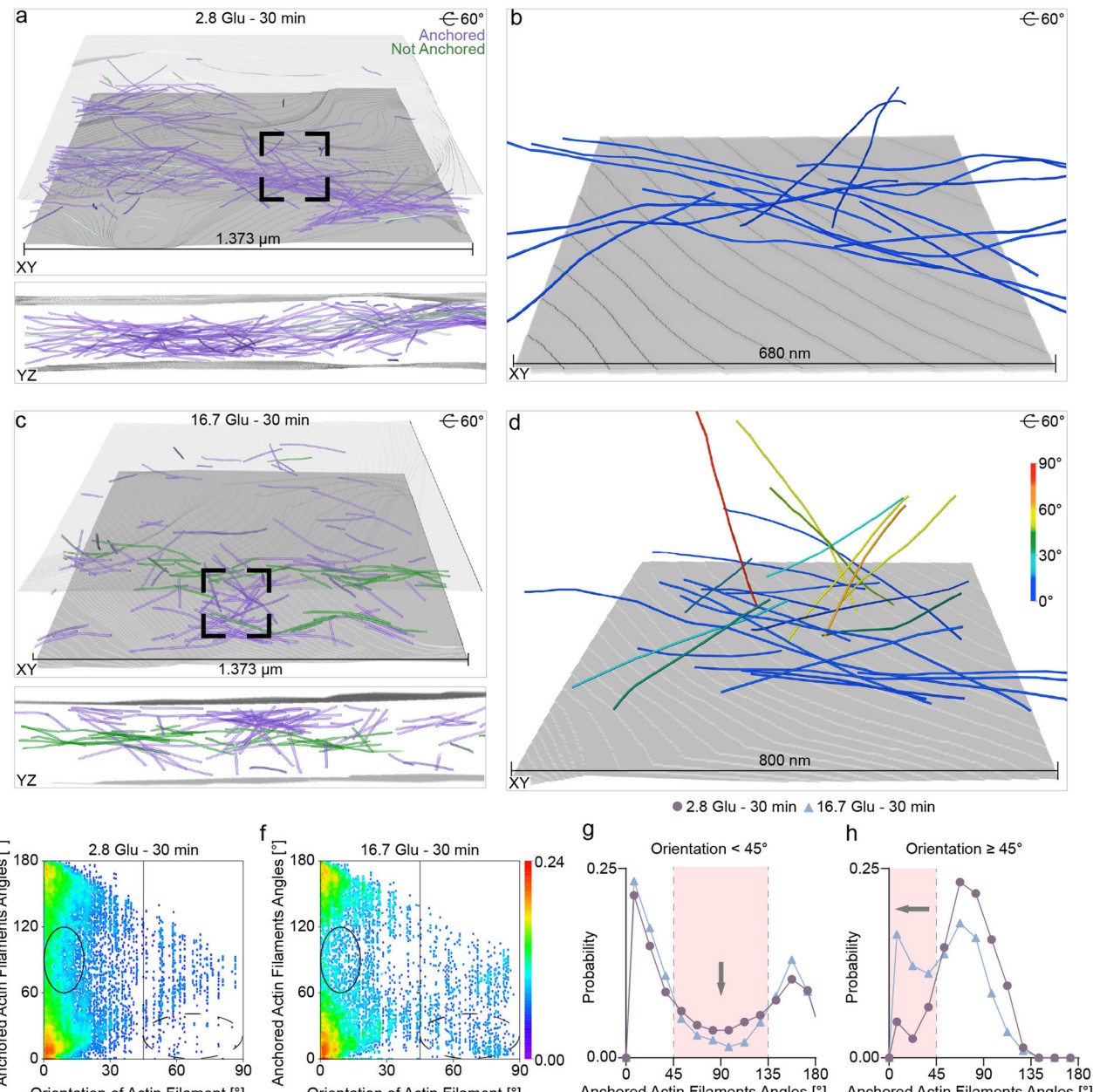

**Fig. 5 | Quantitative analysis of VM-anchored actin filaments at the cell periphery. a, c** 3D visualization of actin filaments (anchored: violet, not anchored: green) before (**a**) and after (**c**) remodeling. **b, d** Zoomed-in views of the actin filaments highlighted in (**a**) and (**c**), represented as a function of their orientation relative to the VM in a blue-to-red color map, before (**b**) and after (**d**) remodeling. **e, f** Density map of the distance between VM-anchored actin filaments as a function of their orientation relative to the VM before (**e**) and after (**f**) remodeling.

**g** Histogram of the angle between VM-anchored actin filaments whose orientation is less than 45° relative to the VM before and after remodeling. Bin size = 15. **h** Histogram of the angles between VM-anchored actin filaments whose orientation relative to the VM is greater than or equal to 45° before and after remodeling. Bin size = 15. Each point on the density map reflects the corresponding density values by calculating kernel density. The tilt axis of the TEM corresponds to the *Y*-axis in the tomograms. Source data are provided as a Source Data file.

oriented to the VM with angles less than 15° have a larger shortest distance to ISGs under the 16.7 Glu – 30 min condition (130–200 nm) compared with basal conditions (70–200 nm) (cyan in Fig. 6j, k). These results indicate a further spatial arrangement of actin filaments relative to ISGs after remodeling compared with basal conditions, which is expected for the subsequent transport and release of ISGs.

The critical role of MT-dependent transport of newly generated ISGs under glucose stimulation has been well established[10,50]. MTs have also been proposed to negatively regulate insulin secretion at the cell periphery under basal conditions, based on a fluorescence microscopy observation of restricted ISG movements during MT stabilization by drugs[17]. Here, we calculated the MT-ISG distance as the distance between the resampled MTs (at 4 nm intervals) and the nearest ISG surface ("Methods"). We observe the highest peak region starting at 20 nm and centered at 200 nm under basal conditions (Fig. 6l), with a significant shift to longer distances under the 16.7 Glu – 30 min condition. This observation provides a structural perspective at the nanoscale on the previously proposed negative regulation of MTs for ISG release under basal conditions[17]. Specifically, MTs "trap" instead of "transport" ISGs, because the transportation requires kinesin-1, which has a size of ~60 nm. Moreover, this effect is significantly reduced under the 16.7 Glu – 30 min condition.

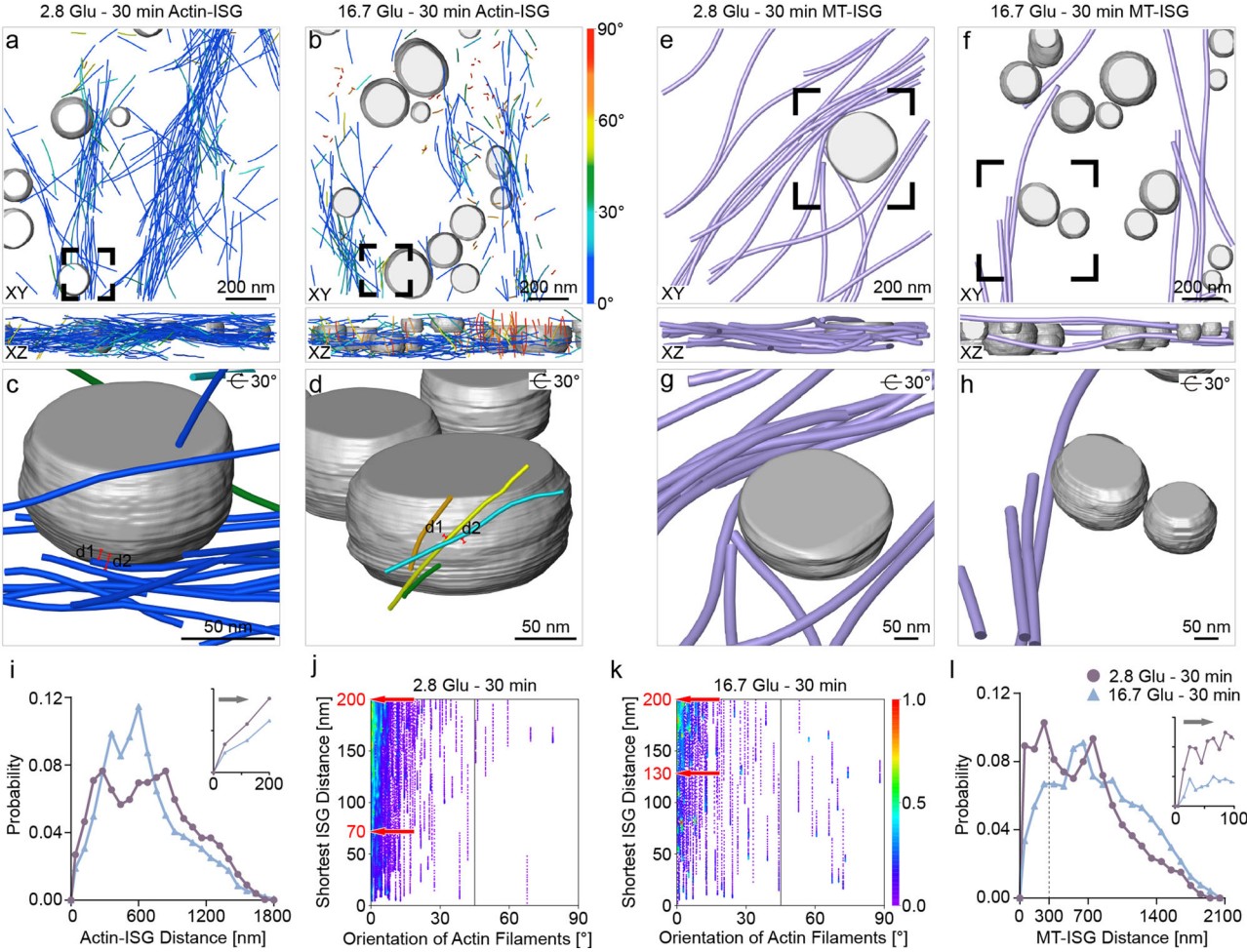

**Fig. 6 | Quantification of the interaction between actin filaments and MTs with ISGs. a**, **b** ISGs (gray) and actin filaments (same color code as in Fig. 4) visualized under basal conditions (**a**) and the 16.7 Glu – 30 min condition (**b**). Results are shown in the XY (top) and XZ (bottom) planes. **c**, **d** Zoomed-in views of actin filaments in the vicinity of ISGs (gray) in positions highlighted by a black box in (**a**) and (**b**), respectively. **c** Example distances d1 = 9 nm, d2 = 17.7 nm. **d** Example distances d1 = 15.4 nm, d2 = 9.1 nm. **e**, **f** ISGs (gray) and MTs (violet) are visualized under basal conditions (**e**) and the 16.7 Glu – 30 min condition (**f**). **g**, **h** Zoomed-in views of the MTs (violet) with ISGs (gray) in positions highlighted by a black box under basal conditions (**g**) and the 16.7 Glu – 30 min condition (**h**). **i** Plot of the distance distribution between actin filaments and ISGs under basal conditions and the 16.7 Glu – 30 min condition. Bin size = 75. **j**, **k** Density maps of the shortest distance between actin filaments and ISGs as a function of their orientation relative to the VM under basal conditions (**j**) and the 16.7 Glu – 30 min condition (**k**). **l** Plot of the distance distribution between MTs and ISGs under basal conditions and the 16.7 Glu – 30 min condition. Bin size = 100. Each point on the density map reflects the corresponding density values by calculating kernel density. The tilt axis of the TEM corresponds to the Y-axis in the tomograms. Source data are provided as a Source Data file.

Lastly, for each actin filament, we computed the distance to the closest MT (shortest MT distance, "Methods") to investigate their interactions (Fig. 7). The total volume of actin filaments and MTs remains constant during GSIS. Actin filaments oriented to the VM at angles of 0–15° tend to be closer to MTs, as indicated by their shortest MT distances in a narrower range under the 16.7 Glu – 30 min condition (20–300 nm) compared to basal conditions (20–800 nm; red to green in Fig. 7e, f). Actin filaments oriented quasi-orthogonally (45–90°) to the VM also tend to be located closer to MTs under the 16.7 Glu – 30 min condition compared to basal conditions (density increase from 1.8% to 13.8%). In summary, actin filaments parallel and quasi-orthogonal to the VM are both closer to MTs under the 16.7 Glu – 30 min condition than under basal conditions.

## Discussion

Actin filaments are known to regulate ISG transport and release under glucose stimulation. Because of the limited resolution of fluorescence microscopy used in previous studies, actin remodeling has been characterized primarily as a "strong distribution" under basal conditions, a "diminished amount" during the first phase, and a "recovery" during the second phase[51]. Here, we applied multimodal imaging to map INS-1E β-cells under basal conditions, as well as under 16.7 mM – 5 min and 16.7 mM – 30 min conditions. At the whole-cell level, we captured the biphasic insulin secretion and actin meshwork organization using ELISA and TIRF (Supplementary Figs. 1 and 2 and Movie 1), measured the G-/F-actin ratio by western blot (Supplementary Fig. 3) and revealed changes in the length and number of junctions in the actin meshwork in randomly selected subsections of the β-cells by SIM (Fig. 1). At the subcellular level, we mapped actin remodeling at the periphery and interior of β-cells by cryo-ET (Figs. 2 and 3). Our work provides the first in situ structure of actin remodeling at the nanoscale, as well as a quantitative analysis of changes in the architecture, alignment and interaction of the actin filaments during GSIS. We establish a model for actin remodeling at the cell periphery during GSIS as follows.

Under basal conditions (Fig. 8, left panel), actin filaments are mostly oriented at angles of 0–45° relative to the VM (Fig. 2), forming bundles that maintain the cell structure (Fig. 4). Actin filaments

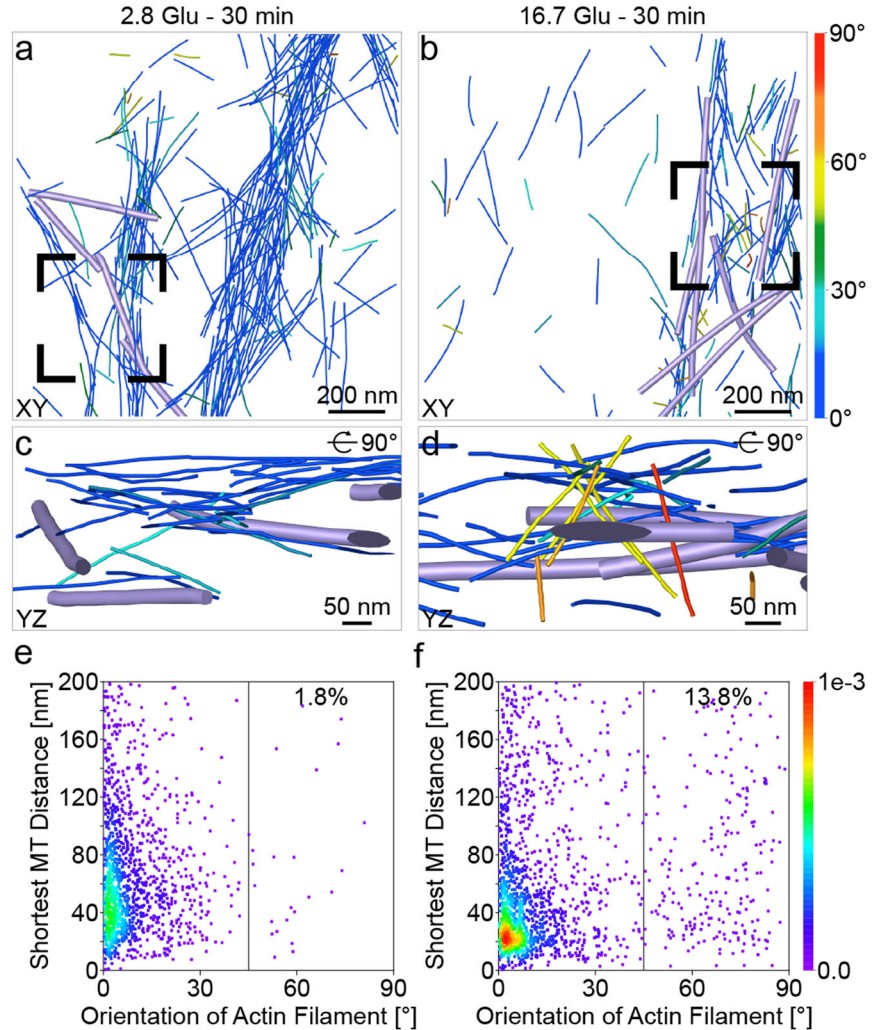

**Fig. 7 | Distance between actin filaments and MTs during GSIS. a, b** Actin filament (same color code as in Fig. 4) and MT (violet) organization shown in the XY plane under basal conditions (**a**) and the 16.7 Glu – 30 min condition (**b**).
**c, d** Zoomed-in views of actin filaments in the vicinity of ISGs (gray) in positions highlighted by a black box in (**a**) and (**b**), respectively. **e, f** Density maps of the shortest distance between actin filaments and MTs under basal conditions (**e**) and the 16.7 Glu – 30 min condition (**f**). Each point on the density map reflects the corresponding density values by calculating kernel density. The tilt axis of the TEM corresponds to the *Y*-axis in the tomograms. Source data are provided as a Source Data file.

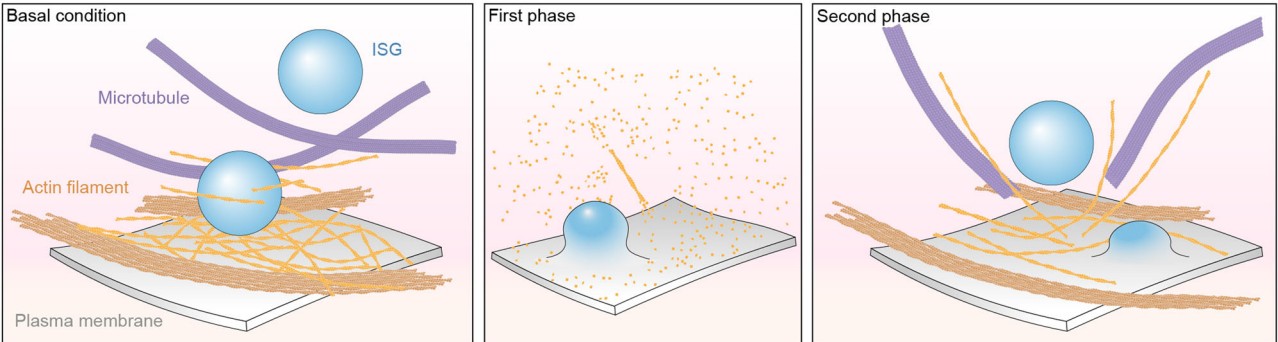

**Fig. 8 | Schematic representation of the model for actin remodeling at the cell periphery during GSIS.** Actin filaments, MTs, and ISGs at the cell periphery during GSIS under the 2.8 Glu – 30 min (basal), 16.7 Glu – 5 min (first phase), and 16.7 Glu – 30 min (second phase) conditions are depicted.

anchored to the VM form a "netlike" architecture that presumably prevents ISGs from approaching the VM (Fig. 5). MTs adopt a close conformation surrounding ISGs, potentially limiting their movement (Fig. 6). Thus, we present a structural perspective on how actin filaments and MTs act as barriers to block the transport and release of ISGs under basal conditions[13,17,52]. Under the 16.7 Glu – 5 min condition (Fig. 8, middle panel), actin filaments are nearly completely depolymerized, MTs and ISGs are almost absent, indicating rapid insulin

secretion. Under the 16.7 Glu – 30 min condition (Fig. 8, right panel), most actin filaments remain organized in bundles (Fig. 4), presumably to fulfill their primary function of maintaining cell structure. Hence, our study presents evidence that actin filaments reconfigure into a new network after repolymerization, leading to three major structural rearrangements that potentially facilitate the transport and release of ISGs from the VM: (i) ~12% of actin filaments reorient themselves quasi-orthogonally to the VM at angles of 45–90° (Fig. 2); (ii) the actin filament network mostly remains as cell-stabilizing bundles but partially reconfigures its architecture into a less compact arrangement (Fig. 4); (iii) actin filaments anchored at the VM reorganize from a "netlike" to a "blooming" architecture, which may be required for transporting ISGs from the cell periphery to the VM (Fig. 5). Additionally, changes in actin filament angles relative to the VM can be hindered by Y15, a focal adhesion kinase inhibitor known to reduce insulin secretion in β-cells (Fig. 4). This suggests that actin angle alterations during remodeling may be regulated by focal adhesion complexes and could be essential for the functional process of insulin secretion in β-cells.

The rearrangements of the actin filament network serve the functional role of regulating insulin secretion. They precede the transport of ISGs by their reorientations quasi-orthogonally to the VM and less compact packing as well as the release of ISGs by the formation of a "blooming" architecture of actin filaments anchored to the VM. Additionally, our analysis reveals changes in interactions among actin filaments, MTs, and ISGs during GSIS. Previous fluorescence studies have led to a proposal that both actin filaments and MTs regulate ISG transport and release by demonstrating limited ISG movements under chemical stabilization of either actin filaments or MTs[13,17]. Here, we show that actin filaments and MTs are further away from ISGs under the 16.7 Glu – 30 min condition compared to the basal conditions (Fig. 6), while being closer to each other (Fig. 7), as expected for the subsequent transport and release of ISGs.

We now discuss the limitations of our work. First, although INS-1E β-cells can recapture biphasic insulin secretion and actin remodeling under glucose stimulation (Supplementary Figs. 1 and 2 and Supplementary Movie 1), their signaling and metabolism pathways still differ from those in primary β-cells[21]. We collected eight tomograms of primary β-cells, contributing to a partial mapping in this regard (Fig. 2). In future work, it would be valuable to map primary β-cells using cryo-ET in healthy and diabetic states to depict variations in their structural and functional mechanisms. Second, we collected tomograms under basal conditions, 16.7 Glu – 5 min and 16.7 Glu – 30 min conditions. Exploring additional time points (e.g., 60 min) holds notable significance in acquiring a better understanding of the functional implications of actin remodeling during GSIS. Third, we only mapped the periphery and interior of the β-cell due to the limitations of cryo-ET in mapping relatively thick cells. Future work should cover more subcellular neighborhoods to obtain a more complete picture of actin remodeling throughout the β-cell. For instance, $Ca^{2+}$ microdomain beneath the VM may be important in the normal regulation of insulin secretion in β-cells[53]. Another future focus should be on the secretion sites near the VM, to gain deeper insights into the secretion mechanism. Indeed, mapping the interactions between actin filaments and other organelles throughout the entire β-cell would provide valuable insights into their importance for ISG positioning, transport and release[54]. Fourth, we analyzed the structure of actin filaments anchored to the VM using a distance threshold. It is known that actin filaments are anchored to the VM via physical connections with focal adhesion complexes at the VM[8]. Cryo-correlative light and electron tomography of fluorescently labeled focal adhesion complexes, followed by subtomogram averaging, could be utilized to detect these complexes and study their interactions with actin filaments during remodeling in greater detail to better understand the molecular mechanisms involved in insulin secretion.

In summary, our study presents the structure and a quantitative analysis of actin remodeling at the nanoscale in both the periphery and interior of β-cells, including changes in the architecture of the actin filament network, the alignment of actin filaments and the interaction among actin filaments, MTs, and ISGs during GSIS. These analyses contribute to a better understanding of actin remodeling and its role in the regulation of biphasic insulin secretion in pancreatic β-cells.

## Methods

### INS-1E and rat primary β-cells culture, treatment, and plasmid transfection

Clonal rat INS-1E β-cells (gifted from P. Maechler's laboratory at the University of Geneva[55]) were cultured in RPMI-1640 GlutaMAX™-1 medium (Thermo Fisher Scientific) containing 11.1 mM glucose, supplemented with 10 mM HEPES (Thermo Fisher Scientific), 5% heat-inactivated fetal bovine serum (FBS) (Thermo Fisher Scientific), 100 U/mL penicillin and streptomycin (Thermo Fisher Scientific), 1 mM sodium pyruvate (Sigma-Aldrich) and 50 µM β-mercaptoethanol (Sigma-Aldrich). Cells detached by 0.25% trypsin (Thermo Fisher Scientific) were seeded at a density of $4 \times 10^4$ cells/cm² on either Falcon culture dishes (Corning) or Poly-L-ornithine (Sigma-Aldrich) coated glass-bottom dishes (ibidi), 6-well plates (Corning), and 12-well plates (Corning), and were grown at 37 °C in 5% $CO_2$. The culture medium was changed 4 days after cell seeding, and cells were passaged every 7 days to maintain the cell line.

The non-immortal rat primary β-cells (Shanghai Zhong Qiao Xin Zhou Biotechnology Co., Ltd.) were cultured in RPMI-1640 Gluta-MAX™-1 medium (Thermo Fisher Scientific) containing 11.1 mM glucose, supplemented with 25 mM HEPES (Thermo Fisher Scientific), 10% heat-inactivated fetal bovine serum (FBS) (Thermo Fisher Scientific), 100 U/mL penicillin and streptomycin (Thermo Fisher Scientific). Cells detached by 0.05% trypsin (Thermo Fisher Scientific) were seeded at a density of $1 \times 10^4$ cells/cm² on either Falcon culture dishes (Corning) or Poly-L-ornithine (Sigma-Aldrich) coated glass-bottom dishes (ibidi) and 6-well plates (Corning), and were grown at 37 °C in 5% $CO_2$.

To achieve different cellular phenotypes under different conditions, both INS-1E β-cells and rat primary β-cells were washed and preincubated for 30 minutes in KREBS solution containing 2.8 mM glucose. Subsequently, the cells were incubated, either for 30 minutes under 2.8 mM glucose or for 5 or 30 minutes under 16.7 mM glucose. For the Y15 treatment, each KREBS solution was supplemented with 1 µM Y15 prior to application on the β-cells.

For cell transfection, INS-1E β-cells were used 48 hours post-cell seeding, and transfected with plasmids of pEGFP-C1.LifeAct-EGFP and pEGFP-N1.NPY-mCherry (modified from RRID: Addgene_58470 and RRID: Addgene_67156, respectively) using lipofectamine 3000 reagent (2 µl lipofectamine and P3000 per 1 mg of plasmid, Thermo Fisher Scientific) to label the ISGs and actin filaments. The transfected INS-1E β-cells were then cultured for 2 days till use.

### Culture and vitrification of INS-1E and rat primary β-cells on EM grid

Quantifoil R 1/4 Au grids with 200 mesh holey carbon film (Quantifoil Micro Tools GmbH) were firstly coated with 0.5% Poly-L-ornithine (Sigma-Aldrich) diluted by phosphate-buffered saline (PBS) (Thermo Fisher Scientific) for 30 minutes. 4–6 coated grids were placed with carbon film upwards in 35-mm Falcon dishes containing 1 mL of complete RPMI 1640 medium, before adding the INS-1E β-cell suspension to reach the density of $4 \times 10^4$ cells/cm². The cells were grown on the grids for 48 hours, and then the grids were blotted from the backside for 10 seconds using a Vitrobot Mark IV (Thermo Fisher Scientific) with 50% humidity, and plunge-frozen in ethane liquid. Subsequently, the vitrified samples were transferred into cryo-EM boxes and stored in liquid nitrogen before FIB-milling or cryo-ET data

collection. For the rat primary β-cell seeding, 4–6 coated grids were placed with carbon film upwards in 35-mm Falcon dishes containing 1 mL of complete tissue culture medium based on the RPMI 1640 medium, before adding the rat primary β-cells suspension to reach the density of $1.5 \times 10^4$ cells/cm². The cells were grown on the grids for 24–48 hours, and then the grids were blotted from the backside for 10 seconds using a Vitrobot Mark IV (Thermo Fisher Scientific) with 50% humidity, and plunge-frozen in ethane liquid. Subsequently, the vitrified samples were transferred into cryo-EM boxes and stored in liquid nitrogen before FIB-milling or cryo-ET data collection.

### Rat primary β-cell isolation and validation

At the company (Shanghai Zhong Qiao Xin Zhou Biotechnology Co., Ltd.), rat islets were isolated and β-cells were dissociated and then captured by Fluorescence-Activated Cell Sorting (FACS). Those freshly sorted β-cells were purchased (PRI-RAT-00136) for subsequent culture and study.

To obtain rat primary β-cells, the entire procedure for isolating islets and obtaining single cells was previously described[56–60]. Briefly, islets were isolated from male Wistar rats (6 weeks old) by collagenase digestion (Worthington) added to 10-fold tissue volume. Digestion was stopped by the addition of ice-cold tissue culture medium (TCM), comprising RPMI-1640 GlutaMAX™-1 medium (Thermo Fisher Scientific) containing 11.1 mM glucose, supplemented with 25 mM HEPES (Thermo Fisher Scientific), 10% heat-inactivated fetal bovine serum (FBS) (Thermo Fisher Scientific), 100 U/mL penicillin and streptomycin (Thermo Fisher Scientific). The digest was filtered through a 500 μm nylon screen. The additional filtering of undigested residues was repeated twice. Subsequently, isolated islets were washed once with precooled PBS. The islets were added with 6 mL of isotonic Percoll and mixed gently by pipetting, an equal volume of PBS slowly was added to form two density gradients, and centrifuged at 2000 rpm horizontally for 20 minutes. Utilizing a pipette, the islets were carefully extracted out between two layers, PBS was added and the resuspension centrifuged at 1000 rpm for 3 minutes. After discarding the supernatant, islets were resuspended in 0.1% trypsin (Thermo Fisher Scientific) and digested for 5–10 minutes. After trypsinization stopped by TCM, the cell suspension was centrifuged at 1000 rpm for 3 minutes and resuspended in 2.8 mM glucose. The cell suspension was then submitted to FACS with 100 mW at 488 nm excitation. The β-cells were collected at 510–550 nm fluorescence emission. The sorted β-cells were cultured on dish (Falcon) at 37 °C in 5% $CO_2$ for further experiments.

For verification of rat primary β-cells, samples were prepared using an indirect immunofluorescence method. Cells were grown on a glass coverslip (Citotest) in a 24-well plate, and stimulated with glucose to acquire three different conditions (basal conditions, the first phase and the second phase of GSIS) as described above. Samples were then washed with 1xPBS three times at room temperature and incubated with 4% paraformaldehyde for 20 minutes, after which cells were immediately washed three times with 1×PBS, followed by permeabilization with 0.5% Triton X-100 in PBS for 20 minutes. Samples were then washed again three times with 1xPBS and cells were blocked in 5% goat serum (Boster Biological Technology) for 30 minutes at room temperature. Next, cells were washed three times with 1×PBS and transferred to the glass slide, then cells were incubated with the primary antibody (Affinity, mouse, 1:100) overnight at 4 °C in a wet box. Samples were then washed with 1xPBS three times, incubated with the secondary antibody (Proteintech, anti-mouse, 1:100) for 60 minutes at room temperature, and washed with 1xPBS three times. Samples were then incubated in Hoechst 33342 working buffer (1 μg/mL, Cell Signaling) for 5 minutes. After this, samples were washed three times with 1xPBS, and the glass coverslip was applied with one drop of Antifade Mountant (SouthernBiotech). Finally, the samples were stored at −20 °C in the dark. Immunofluorescence imaging was performed on an Olympus BX53 (0L48573) using a UPlanFL N 40× 0.75 objective. Fluorescence channels were DAPI (Ex 354–364 nm, Em 456–466 nm), and Cy3 (Ex 545–555 nm, Em 565–575 nm) (Supplementary Fig. 5).

### Live-cell fluorescence imaging

To record the INS-1E and rat primary live-cell images during the time of GSIS, cells were grown on 35 mm glass-bottom dishes (ibid), transfected with plasmids to label ISGs and actin filaments, and stimulated with glucose as described above. The movies were acquired using wide-field TIRF mode under light microscope system at 37 °C in 5% $CO_2$. This light Microscopy was performed on a Nikon Ti2-E with TIRF equipped with a Prime 95B Scientific CMOS using an Apo TIRF 60× Oil 1.49 objective and NIS-Elements AR 64-bit (v5.21.00, Nikon). Fluorescence channels were Hoechst 33342 (Ex 345–355 nm, Em 450–460 nm), EGFP (Ex 483–493 nm, Em 502–512 nm) and mCherry (Ex 582–592 nm, Em 605–615 nm). Finally, the resolution of TIRF data is x-122.2 nm, y-122.2 nm. TIRF data was analyzed using Huygens (v19.4, Scientific Volume Imaging), ImageJ (v1.53f51) and homemade Python scripts (v3.7.7).

### Western blot

The ratio of G- and F-actin was performed following the manufacturer's protocols (Cytoskeleton). Protein samples were separated by SDS-PAGE and transferred to PVDF membrane (Thermo Fisher) with known amounts of actin used for quantitation in G-/F-actin ratio experiments. Membranes were blocked in 5% skim milk powder in tris buffered saline (TBS, 50 mM Tris pH 7.5, 150 mM NaCl) with 0.1% Tween 20 (Merck). Incubate with the primary antibody (rabbit) provided in the kit overnight at 4 °C. Then incubate with the secondary antibody (Abcam, anti-rabbit, 1:5000). Finally, the membrane was exposed after incubation in a dark environment with ECL substrate for 1 minute (Bio-rad) with Image Lab (v5.2, Bio-Rad). Western blot data were analyzed using ImageJ (v1.53f51) and GraphPad Prism (v9.4.1, GraphPad Software).

### Insulin secretion level

Insulin secretion was quantified as the cumulative insulin release across each treatment period. The INS-1E β-cells and rat primary β-cells were prepared in 12-well plates (Corning) and exposed to KREBS solution containing 2.8 mM glucose for 30 minutes as the preincubation condition. Subsequently, the cells were incubated: (i) for 30 minutes under 2.8 mM glucose, (ii) for 5 minutes under 16.7 mM glucose, and (iii) for 30 minutes under 16.7 mM glucose conditions, respectively. The supernatant was aspirated at the respective end time point for each condition. Each sampling well was treated as an independent sample. After the supernatant was removed, cells were washed with 1xPBS once and detached by trypsin. Following centrifuge at 1000 rpm for 3 minutes, cells were resuspended in 1 mL of complete culture medium and then counted. Insulin levels in the supernatant were measured using an insulin ELISA kit (Mercodia). The protein content was subsequently evaluated using Flexstation 3.0 (Molecular Devices) with SoftMax Pro (v5.4.5.000, Molecular Devices). Finally, insulin secretion data were analyzed using GraphPad Prism (v9.4.1, GraphPad Software).

### Super-resolution fluorescence microscopy of fixed INS-1E β-cells

To prepare the fixed INS-1E β-cell samples, cells were grown on a glass coverslip (ibidi), transfected with plasmids to label ISGs and actin filaments, and stimulated with glucose to acquire three different conditions (2.8 Glu – 30 min, 16.7 Glu – 5 min and 16.7 Glu – 30 min) as described above. Then the samples were washed with 1xPBS three times at room temperature, and incubated with 4% paraformaldehyde for 20 minutes. Sequentially the cells were immediately washed three times with 1×PBS and incubated in 2 mL of Hoechst 33342 working buffer (1 μg/mL, Cell Signaling) for 5 minutes. After that, samples were

washed three times with 1xPBS, and the glass coverslip was applied with one drop of ProLong™ Glass Antifade Mountant (Thermo Fisher Scientific) and transferred onto the glass slide. Finally, the samples were kept at room temperature for 48 hours and later stored at −20 °C in the dark. The super-resolution fluorescence images were collected in SIM (structured illumination microscopy) mode on Zeiss Elyra 7 with Lattice SIM, equipped with a PCO edge 4.2 sCMOS camera using a Plan-Apochromat 63x/1.4 Oil DIC M27 objective and Zen 64-bit (3.0 SR FP1 black, Carl Zeiss). Fluorescence channels were Hoechst 33342 (Ex 345–355 nm, Em 450–460 nm), EGFP (Ex 483–493 nm, Em 502–512 nm) and mCherry (Ex 582–592 nm, Em 605–615 nm). Finally, the resolution of the SIM data is x−31.3 nm, y−31.3 nm, and z−90.9 nm. SIM data was analyzed using ImageJ (v1.53f51) and GraphPad Prism (v9.4.1, GraphPad Software).

## Fluorescence image analysis

Labeling of the actin meshwork was accomplished using "AnalyzeSkeleton" with ImageJ (v1.53f51) as follows: (1) the actin was first subjected to filtering using the "Threshold-default" tool; (2) the actin was skeletonized using the "skeletonize(2D/3D)" tool; (3) key features of the actin, such as coordinates, total length and network junctions, were determined using the "AnalyzeSkeleton" tool[27,28]. Here, we exclusively considered the actin meshwork with lengths exceeding one pixel. To quantify the total length and network junction of actin meshwork from the super-resolution fluorescence images of INS-1E β-cells (Fig. 1), a total of 72 subsections were selected using the "Cut" tool within Adobe Photoshop CC 19.1.19 and labeled utilizing the "AnalyzeSkeleton" function. For each condition, a total of 24 subsections were analyzed from all six cell images, with each subsection covering an area of 0.97 μm². To determine the angle between the actin meshwork and the ventral membrane (VM, the plasma membrane in direct contact with the culture surface) (Fig. 4), the actin meshwork positioned within a 300 nm proximity to the VM was extracted using the "AnalyzeSkeleton" function. This distance threshold aligns with the cell height seen in cryo-electron tomograms (Supplementary Fig. 10a). For each condition, this process was performed for all six or ten super-resolution fluorescence images of INS-1E β-cells in the absence or presence of Y15 treatments, respectively. This angle was calculated as the angle between the vectors connecting two end-points of the actin meshwork and the x-y plane of the VM.

## Cryo-FIB milling

Cryo-FIB milling was carried out following similar procedures as previously described[61], using a dualbeam Aquilos 2 Cryo-FIB microscope (Thermo Fisher Scientific). EM grids with frozen cells were clipped into autogrid support rings with a cutout region to facilitate shallow-angle cryo-FIB milling. The Autogrids were mounted onto the cryo-FIB AutoGrid shuttle and transferred to the cryo-stage at liquid nitrogen temperatures. Before the milling procedure, grids were sputter-coated with platinum for 30 seconds (10 mA) and subsequently sputter-coated with organometallic platinum using the gas injection system (GIS, Thermo Fisher Scientific) for 8 seconds to improve conductivity and remove artifacts. Samples were milled by gallium ion beam at 30 kV with a stage tilting angle of 17–19° to generate 10–12 μm wide lamellas, the initial rough milling was done under 0.5 nA high currents, then the current was gradually decreased in a stepwise manner to 30 pA for fine milling and final polishing. Electron beam at 2 kV/13 pA or 5 kV/25 pA was used for SEM imaging during the milling process. In total, 18 lamellas from randomly chosen cells were used for the following cryo-ET data collection.

## Cryo-ET acquisition and reconstruction

The grids containing either vitrified INS-1E β-cells or lamella samples produced by cryo-FIB milling were loaded in a Titan Krios G3 (Thermo Fisher Scientific) with TEM User Interface (v2.15.3, Thermo Fisher

Scientific) and SerialEM (v3.8.0 beta, University of Colorado) or a Krios G4 TEM (Thermo Fisher Scientific) with TEM User Interface (v3.9.1, Thermo Fisher Scientific) and Tomography (v5.8.0.3166REL, Thermo Fisher Scientific). The Titan Krios G3 TEM was equipped with a 300-kV field-emission gun, a post-column energy filter (Gatan), and a 5760 × 4092 K3 Summit direct electron detector (Gatan), operated using SerialEM[62]. Low-magnification images were captured at 3600×. High-magnification tilt series were recorded in counting mode at 26,000x (calibrated pixel size of 0.3353 nm). The tilt axis corresponds to the Y-axis of the tomograms. The Krios G4 TEM (Thermo Fisher Scientific) was equipped with a 300-kV C-FEG field-emission gun, a post-column energy filter (Selectris), and a 4096 × 4096 direct electron detector (Falcon4), operated using Tomography (Thermo Fisher Scientific). Low-magnification images were captured at 5600×. High-magnification tilt series were recorded in counting mode at 42,000x (calibrated pixel size of 0.3028 nm). The tilt axis corresponds to the X-axis of the tomograms. For cell periphery samples, tilt series were collected with ±60° tilt range, 2° step. 24 tilt-series, collected from 16 different cells were collected in total. For lamella samples, a similar scheme covering 120° with 2° increments starting from -10° compensating the pre-tilt of lamella was applied. 18 tilt-series, collected from 18 different cells, were collected in total. All 42 tomograms were recorded at a total dose of 110–140 e-/Å² using a dose-symmetric tilt scheme[63] and a target defocus range of −3 to −7 μm (Supplementary Tables 2 and 3).

Data preprocessing including motion correction by MotionCor2[64] (v1.5.0, UCSF) and the dose-filtering step[65] was performed using the TOMOMAN package which could execute on MATLAB (R2019b, MathWorks) software (https://github.com/williamnwan/TOMOMAN). Tilt-series were aligned by patch tracking in IMOD[66] (v4.9.12, University of Colorado) and reconstructed to 4x binned tomograms (with a pixel size of 13.4 Å and 12.1 Å for cell periphery and lamella (cell interior), respectively) using weighted-back projection. Tomograms were then denoised by the cryoCARE algorithm (v0.2.2, https://github.com/juglab/cryoCARE_T2T) for better segmentation and visualization.

## Filament and membrane segmentation

The above mentioned 4× binned tomograms were used for the segmentation. Correlative volumes of membrane positions were detected and generated automatically using tomosegmemtv[38] and imported to Amira software (v2019.2, Thermo Fisher Scientific) for manual refinement and segmentation. Actin filaments and MTs were traced automatically in Amira software using an automated segmentation algorithm, which uses a cylinder as a template[37] and implemented in the X-Tracing extension. The cylindrical templates were generated with a length of 60 nm or 100 nm and diameters of 8 nm or 15 nm for the actin filaments or MTs, respectively. After the automated segmentation algorithm, the segmented results were manually checked using the tomogram. The identification of the VM positions within the tomogram is achieved through a manual segmentation process, whereby both intracellular and extracellular regions are recognized to create a VM mask.

## Cryo-ET data analysis

Cryo-ET data analysis was performed using in-house Python scripts using coordinates of actin filaments and MTs, VM mask volumes, and ISG mask volumes, which were exported from Amira software. First, we adjusted the coordinates based on the offset of each tomogram. Next, we resampled actin filaments and MTs with 4 nm intervals; this interval has been previously shown to provide a reasonable fit in other analyses[32]. The Actin-Actin distance represents the distance between the centerline of two actin filament segments, which was calculated as the Euclidean distance between each resampled point on the actin filament and its nearest resampled points on all other actin filaments (Fig. 4). The actin filament angles represent the relative orientations

between each two actin filaments. This angle was calculated as the angle between the vectors formed by two end points of the actin filament and the vectors formed by two end points of all other filaments (Fig. 4). The anchored actin filament angles represent the relative orientations between two nearby actin filaments anchored to the VM, the plasma membrane in direct contact with the carbon film. For each anchored actin filament, the neighboring filaments are identified if: (1) they were anchored to the same VM, and (2) the distance between their end points near the VM was less than 120 nm, which is twice the length of the actin filament segmentation unit (60 nm). The angle was then calculated between each anchored filament vector extending from the end point near the VM to the end point further away from it and all other nearby anchored filament vectors (Fig. 5). The Actin-ISG distance was calculated as the Euclidean distances of each resampled point of actin filaments to the nearest ISG surface (Fig. 6). The MT-ISG distance was calculated in the same way as the Actin-ISG distance (Fig. 6). The shortest MT distance of actin filament was calculated as the Euclidean distance between each resampled point on the actin filament and its nearest resampled points on all MTs (Fig. 7).

All statistical analysis was carried out using GraphPad Prism (v9.4.1) and OriginPro (v9.8.0.200). Statistical significance tests for the total length and network junction of actin meshwork, the analysis of both cell periphery and interior of the cell ratio of actin filament-vol/tomo-vol and MT-vol/tomo-vol, and orientation of actin filaments at the cell interior were calculated using one-way ANOVA test, followed by Tukey's multiple comparisons tests. Statistical significance tests for the orientation of peripheral actin filaments, the average cell height, the number of anchored actin filaments, and the percentage of anchored actin filaments by total actin filaments were calculated using an *F*-test and *t*-test. Values are reported as the mean values. The error bars represent SEMs unless otherwise stated.

**Reporting summary**
Further information on research design is available in the Nature Portfolio Reporting Summary linked to this article.

## Data availability
The data generated in this study are provided in the Supplementary Information/Source Data file. The tomograms used in this study are available in Electron Microscopy Data Bank (EMDB) under accession codes: EMD-35841, EMD-35809, EMD-35842, EMD-35843, EMD-35844, EMD-35845, EMD-35846, EMD-35847, EMD-35855, EMD-35840, EMD-35856, EMD-35857, EMD-35858, EMD-35859, EMD-35860, EMD-35861, EMD-35849, EMD-35850, EMD-35851, EMD-35852, EMD-35839, EMD-35848, EMD-35853, EMD-35854, EMD-35935, EMD-35874, EMD-35896, EMD-35875, EMD-35876, EMD-35885, EMD-35886, EMD-35887, EMD-35897, EMD-35889, EMD-35890, EMD-35936, EMD-35891, EMD-35892, EMD-35893, EMD-35894, EMD-35937, EMD-35895, EMD-37278, EMD-37279, EMD-37280, EMD-37281, EMD-37282, EMD-37283, EMD-37284, EMD-37285. Source data are provided with this paper.

## Code availability
Python scripts used for actin filament and MT resampling, and to quantify Actin-Actin distances, actin filaments angles, anchored actin filament angles, Actin-ISG distances, MT-ISG distances, and shortest MT distances of actin filaments are available at https://github.com/SaliLab-SH/iPA/tree/ActinAnalysis.

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

## Acknowledgements

We thank the members of the Wolfgang Baumeister's Department at Max Planck Institute of Biochemistry, in particular, Sven Klumpe for the cryo-FIB technical support, Jonathan Schneider for the actin filament

segmentation and Antonio Martinez-Sanchez for computational assistance. We thank the Imaging Facility at Max Planck Institute of Biochemistry for fluorescence imaging support. We thank HPC (High-Performance Computing) at ShanghaiTech University and Max Planck Institute of Biochemistry for their computational assistance. We thank Bio-EM Facility, Imaging Core, Purification Core and Mammalian Core at ShanghaiTech University for wet lab support, machine maintenance and technical assistance. We would like to thank the Molecular Imaging Core Facility (MICF), School of Life Science and Technology, ShanghaiTech University, for our SIM and TIRF, and we are grateful to Xiaoming Li and Ziwei Yang for their help in taking images. We thank Pierre Maechler's lab at the University of Geneva for providing the INS-1E β-cell line. We appreciate the discussions with Valentina Loconte, Luca Zinzula, Xianjun Zhang, Lu Zhuang, Yan Liu and all members of the Pancreatic β-Cell Consortium (http://www.pbcconsortium.org). We thank Shanghai Frontiers Science Center for Biomacromolecules and Precision Medicine at ShanghaiTech University, Shanghai Municipal Government, and ShanghaiTech University for supporting our research. We also acknowledge funding from National Institutes of Health of USA (NIGMS R01GM083960 and P41GM109824 to A.S.).

## Author contributions

W.B., A.S. and L.S. designed the project. W.B., A.S., M.J. and L.S. supervised the project. W.L. conducted cell culture and seeding, plasmid concentration and transfection, Western Blot, TIRF microscopy, SIM, cryo-FIB milling, and cryo-ET experiments. W.L., A.L. and B.Y. performed data analysis. X.L. monitored the INS-1E β-cell line using biochemical assays. X.Z., K.L.W. and R.C.S. provided critical guidance. W.L. and L.S. wrote the manuscript with edits from A.S. and M.J. and inputs from all other authors.

## Competing interests

The authors declare no competing interests.
