## [Peer Review File · Nature Communications]

In situ structure of actin remodeling during glucose-stimulated insulin secretion using cryo-electron tomographyEditorial Note: This manuscript has been previously reviewed at another journal that is not operating a transparent peer review scheme. This document only contains reviewer comments and rebuttal letters for versions considered at *Nature Communications*.

REVIEWER COMMENTS

Reviewer #1 (Remarks to the Author):

The authors have satisfactorily addressed my previous comments.

Reviewer #2 (Remarks to the Author):

Major concerns:

1. INS-1E cells should not be considered full mimics of primary islet beta cells in terms of insulin release phases. Hence, the authors should make corrections throughout the manuscript to redact language regarding phases in text regarding data from INS-1E cells (e.g. page 8, line 251-256—change “first phase” to “5 min”; change “second phase” to “30 min”, more precisely denoting the timepoints at which the data were captured (e.g. 5 min, 30 min).
2. Why are there so very few ISGs in the primary rat beta cell images? Primary islet beta cells are well-described in the literature as containing many ISGs, generally more densely packed with ISGs than seen in clonal cells. Please replace current images with those more representative of healthy primary beta cells.
3. Methods and validations concerning the rat primary islet beta cells are missing. For example, the type of rat, male/female, age and weight, the method of isolation of the islets from the rat, and the method of isolation of the beta cells from the islets is missing. Importantly, how were the beta cells validated as beta cells? Given the atypical phenotype of the primary islet beta cells mentioned in concern #2 above, proof of validation that the cells imaged were beta cells is required, and that the cells purchased were capable of glucose-induced biphasic insulin release.
4. The absence of demonstration of glucose-induced insulin release at the 5 min timepoint elicits concern about the cell quality or conditions used. Detection of insulin release within the first 5 min is routine whilst using healthy cells.

Reviewer #3 (Remarks to the Author):

The authors should confirm to the reviewer that the analysis is indeed related to the orientation of filaments to the ventral membrane, where the ventral membrane is oriented parallel to the carbon film. Once this is confirmed, the authors should accurately revise their definition from "plasma membrane (PM)" to "Ventral Membrane (VM)" throughout the paper. However, if the analysis pertains to the orientation of filaments to the plasma membrane (and at the leading edge), as currently written, the additional analysis provided by the authors in the revision suggests that complete quantification is not supported.

Response to reviewers

Reviewer #2 (Remarks to the Author):

Major concerns:

1. INS-1E cells should not be considered full mimics of primary islet beta cells in terms of insulin release phases. Hence, the authors should make corrections throughout the manuscript to redact language regarding phases in text regarding data from INS-1E cells (e.g. page 8, line 251-256—change “first phase” to “5 min”; change “second phase” to “30 min”, more precisely denoting the timepoints at which the data were captured (e.g. 5 min, 30 min).

Response:

We thank the reviewer for the valuable comments and suggestions. We have corrected the reference to “first phase” to “16.7 Glu - 5 min”, and to “second phase” to “16.7 Glu - 30 min” regarding INS-1E cell data throughout the manuscript and supplementary information (highlighted in yellow).

2. Why are there so very few ISGs in the primary rat beta cell images? Primary islet beta cells are well-described in the literature as containing many ISGs, generally more densely packed with ISGs than seen in clonal cells. Please replace current images with those more representative of healthy primary beta cells.

Response:

We thank the reviewer for the comments and suggestions. We agree that primary β -cells are generally more densely packed with ISGs than clonal cells. Unfortunately, cryo-ET is associated with certain limitations, which restrict us to capturing relatively small cellular regions, approximately $1.53 \mu\text{m}^2$, representing just about 1% of the cytosol volume. In tomograms of INS-1E β -cells, variations in ISG density have been observed in previous studies¹. Depending on the specific area captured during data collection, some regions in the tomograms show few ISGs, while others reveal several. During our data collection, we aimed to observe actin filaments near the cell's ventral membrane in order to obtain sufficient statistics for subsequent data analysis. We have therefore not concentrated on regions condensed with ISGs, where few actin filaments can be observed.

1. Zhang, X. et al. Visualizing insulin vesicle neighborhoods in β cells by cryo-electron tomography. *Science Advances* 6, eabc8258 (2020).

We agree with the reviewer's suggestion that a more representative tomogram of healthy primary β -cells is required. We have updated Figure 2, replacing the current

tomogram with one featuring a complete ISG under the 2.8 mM - 30 min condition, as follows:

Figure 2. Quantitative analysis of actin filament and MT orientations relative to the VM at the cell periphery during GSIS. a, b, c Slices through INS-1E β -cell tomograms under basal conditions (2.8 Glu - 30 min; **a**), in the 16.7 Glu - 5 min, **b**, and in the 16.7 Glu - 30 min, **c**. Arrows indicate actin filaments (orange), MTs (violet), ISGs (blue), lysosomes (silver), and ribosomes (white). **d, e, f** Respective segmentations with the same color code. Non-ISG membranes are shown in gray. **g, h** Volume ratio of actin filaments (**g**) or MTs (**h**) to the tomogram in each INS-1E β -cell tomogram under different conditions.

i Orientation of actin filaments relative to the VM under different conditions in INS-1E β -cell tomograms. j, k Slices through rat primary β -cell tomograms under basal conditions (j), and in the 16.7 Glu - 30 min (k). l Orientation of actin filaments relative to the VM under different conditions in rat primary β -cell tomograms. * indicates $p < 0.05$. ** indicates $p < 0.01$. *** indicates $p < 0.001$. **** indicates $p < 0.0001$ by one-way ANOVA. $n = 24$ tomograms, corresponding to eight tomograms for each condition from three biologically independent experiments. $n = 8$ tomograms in rat primary β -cells, corresponding to four tomograms for each condition from three biologically independent experiments. The tilt axis of the transmission electron microscope (TEM) corresponds to the Y-axis in the tomograms under basal condition and 16.7 Glu - 30 min (Supplementary Table 3). As shown in Supplementary Fig. 8, actin filament orientations are distributed over the full angular range, including at orientations close to the XZ plane, highlighting the differences between the patterns of the two experimental conditions.

In addition, we have included a low-magnification view of the periphery of rat primary β -cells in Supplementary Fig. 6. In both the INS-1E β -cell tomogram and the rat primary β -cell tomogram, multiple ISGs can be readily identified at the cell periphery.

Supplementary Fig. 5 Low magnification view of the periphery of vitrified INS-1E β -cells and rat primary β -cells. a An INS-1E β -cell vitrified on EM grids and imaged by cryo-TEM at 3600x magnification. The edge of the cell is delineated on the map with a black line. The cell periphery (0-6 μ m from the plasma membrane) is indicated on the map. b Slice through a cryo-electron tomogram collected at the position indicated by the white box in a. c A rat primary β -cell vitrified on EM grids and imaged by cryo-TEM at 4800x magnification. The edge of the cell is delineated on the map with a black line. The cell periphery

(0-6 μm from the plasma membrane) is indicated on the map. **d** Slice through a cryo-electron tomogram collected at the position indicated by the white box in **c**. Insulin secretory granules were marked by blue arrow. Ice were marked by white star.

3. Methods and validations concerning the rat primary islet beta cells are missing. For example, the type of rat, male/female, age and weight, the method of isolation of the islets from the rat, and the method of isolation of the beta cells from the islets is missing. Importantly, how were the beta cells validated as beta cells? Given the atypical phenotype of the primary islet beta cells mentioned in concern #2 above, proof of validation that the cells imaged were beta cells is required, and that the cells purchased were capable of glucose-induced biphasic insulin release.

Response:

We thank the reviewer for the comments and suggestions. We have updated the details regarding the methods of rat primary β -cell isolation and validation in the **Methods** section as follows (page 20, line 587-617): "To obtain rat primary β -cells, islets were isolated from male Wistar rats (6 weeks old) via collagenase digestion, as described previously. Subsequently, isolated islets were washed once with precooled PBS, resuspended in ice-cold collagenase digestion solution (Worthington, added with ten times volume of tissue). Digestion was stopped by the addition of ice-cold tissue culture medium (TCM), comprising RPMI-1640 GlutaMAX™-1 medium (Thermo Fisher Scientific) containing 11 mM glucose, supplemented with 25 mM HEPES (Thermo Fisher Scientific), 10% heat-inactivated fetal bovine serum (FBS) (Thermo Fisher Scientific), 100 U/ml penicillin and streptomycin (Thermo Fisher Scientific). Finally, dispersed rat primary β -cells were resuspended in TCM after centrifugation and used for further experiments.

For verification of rat primary β -cells, samples were prepared using an indirect immunofluorescence method. Cells were grown on a glass coverslip (Citotest) in a 24-well plate, and stimulated with glucose to acquire three different conditions (basal conditions, the first phase and the second phase of GSIS) as described above. Samples were then washed with 1xPBS three times at room temperature and incubated with 4% paraformaldehyde for 20 min, after which cells were immediately washed three times with 1xPBS, followed by permeabilization with 0.5% Triton X-100 in PBS for 20 min. Samples were then washed again three times with 1xPBS and cells were blocked in 5% goat serum (Boster Biological Technology) for 30 min at room temperature. Next, cells were washed three times with 1xPBS and transferred to the glass slide, then cells were incubated with the primary antibody (Affinity, mouse, 1:100) overnight at 4°C in a wet box. Samples were then washed with 1xPBS three times, incubated with the secondary antibody (Proteintech, anti-mouse, 1:100) for 60 min at room temperature, and washed with 1xPBS three times. Samples were then incubated in Hoechst 33342 working buffer (1 $\mu\text{g}/\text{mL}$, Cell Signaling) for 5 min. After this, samples were washed three times with 1xPBS, and the glass coverslip was applied with one drop of Antifade Mountant (SouthernBiotech). Finally, the samples were stored at -

20°C in the dark. Immunofluorescence imaging was performed on an Olympus BX53 (OL48573) using a UPlanFL N 40x 0.75 objective. Fluorescence channels were DAPI (Ex 354-364 nm, Em 456-466 nm), and Cy3 (Ex 545-555 nm, Em 565-575 nm). (Supplementary Fig. 5)”

We have included the validation of rat primary beta cells by immunofluorescence microscopy in Supplementary Fig. 5. In these immunofluorescence microscopy images, the nucleus and insulin are visualized by the red and blue channels, respectively.

Supplementary Fig. 5 Validation of rat primary β -cells using immunofluorescence microscopy. Rat primary β -cells labeled with Cy3 for insulin secretory granules (red, a) and DAPI for nucleus (blue, b) were imaged using widefield microscopy. These two channels were merged in c.

To further confirm the capability of rat primary β -cells for glucose-stimulated biphasic insulin secretion, we performed ELISA (new data, Supplementary Figure 1, see below) and TIRF (new data, Supplementary Figure 2, see below). We have updated the text in manuscript (page 6, line 208-210):” After confirming that rat primary β -cells exhibit glucose-stimulated biphasic insulin secretion (Supplementary Fig. 1, Supplementary Fig. 2, and Supplementary Movie 2), we performed cryo-ET under two conditions: 2.8 Glu - 30 min and 16.7 Glu - 30 min.”

Supplementary Fig. 1 Insulin secretion levels under different conditions in both INS-1E β -cells and rat primary β -cells. a Insulin secretion from INS-1E β -cells were measured under different conditions by ELISA. n = 4. b Insulin secretion from rat primary β -cells were measured under different conditions by ELISA. n = 4.

Supplementary Fig. 2 Quantification of insulin secretory granules in in both INS-1E β -cells and rat primary β -cells, using live-cell total internal reflection fluorescence (TIRF) during GSIS. a INS-1E β -cells labeled with NPY for insulin secretory granules (red) were imaged in TIRF. **b** Rat primary β -cells labeled with NPY for insulin secretory granules (red) were imaged in TIRF. Both cells were starved in a 2.8 mM glucose KREB solution for 30 min, and then stimulated in a 16.7 mM glucose solution at 0 min. The fluorescence intensity of insulin secretory granules was recorded over time in live cells. A total of six TIRF images were collected from individual INS-1E β -cells and rat primary β -cells, respectively, all of which originated from biologically independent experiments. Signal intensities of insulin secretory granules were normalized by the maximum of the averaged intensity over six images.

4. The absence of demonstration of glucose-induced insulin release at the 5 min timepoint elicits concern about the cell quality or conditions used. Detection of insulin release within the first 5 min is routine whilst using healthy cells.

Response:

To verify the capacity of rat primary β -cells for glucose-induced biphasic insulin release, we conducted ELISA (new data, Supplementary Figure 1, see below) and TIRF (new data, Supplementary Figure 2, see below). In rat primary β -cells, insulin release was measured at 2.3 ± 0.3 , 1.7 ± 0.2 , and 4.4 ± 0.2 ng per million cells under 2.8 Glu - 30 min, 16.7 Glu - 5 min, and 16.7 Glu - 30 min conditions, respectively. These secretion patterns align well with results obtained from isolated islets^{1,2}. The amount of insulin release differs between rat primary β -cells and INS-1E β -cells; however, the changes in insulin secretion under glucose stimulation align within a similar range, as previously documented³. Together with TIRF experiments (new data, Supplementary Figure 2, see below), we confirm that both INS-1E β -cells and rat primary β -cells demonstrate glucose-stimulated biphasic insulin secretion.

1. De Paula, F. M. M., Boschero, A. C., Carneiro, E. M., Bosqueiro, J. R. & Rafacho, A. Insulin signaling proteins in pancreatic islets of insulin-resistant rats induced by glucocorticoid. *Biol. Res.* 44, 251–257 (2011).
2. Purrello, F., Vetri, M., Gatta, C., Gullo, D. & Vigneri, R. Effects of high glucose on insulin secretion by isolated rat islets and purified β -cells and possible role of glycosylation. *Diabetes* 38, 1417–1422 (1989).
3. Merglen, A. et al. Glucose sensitivity and metabolism-secretion coupling studied during two-year continuous culture in INS-1E insulinoma cells. *Endocrinology* 145, 667–678 (2004).

We have updated the text in manuscript (page 6, line 208-210): "After confirming that rat primary β -cells exhibit glucose-stimulated biphasic insulin secretion (Supplementary Fig. 1, Supplementary Fig. 2, and Supplementary Movie 2), we performed cryo-ET under two conditions: 2.8 Glu - 30 min and 16.7 Glu - 30 min."

Supplementary Fig. 1 Insulin secretion levels under different conditions in both INS-1E β -cells and rat primary β -cells. a Insulin secretion from INS-1E β -cells were measured under different conditions by ELISA. n = 4. **b** Insulin secretion from rat primary β -cells were measured under different conditions by ELISA. n = 4.

Supplementary Fig. 2 Quantification of insulin secretory granules in in both INS-1E β -cells and rat primary β -cells, using live-cell total internal reflection fluorescence (TIRF) during GSIS. a INS-1E β -cells labeled with NPY for insulin secretory granules (red) were imaged in TIRF. **b** Rat primary β -cells labeled with NPY for insulin secretory granules (red) were imaged in TIRF. Both cells were starved in a 2.8 mM glucose KREB solution for 30 min, and then stimulated in a 16.7 mM glucose solution at 0 min. The fluorescence intensity of insulin secretory granules was recorded over time in live cells. A total of six TIRF images were collected from individual INS-1E β -cells and rat primary β -cells, respectively, all of which originated from biologically independent experiments. Signal intensities of insulin secretory granules were normalized by the maximum of the averaged intensity over six images.

Reviewer #3 (Remarks to the Author):

The authors should confirm to the reviewer that the analysis is indeed related to the orientation of filaments to the ventral membrane, where the ventral membrane is oriented parallel to the carbon film. Once this is confirmed, the authors should accurately revise their definition from "plasma membrane (PM)" to "Ventral Membrane (VM)" throughout the paper. However, if the analysis pertains to the orientation of filaments to the plasma membrane (and at the leading edge), as currently written, the additional analysis provided by the authors in the revision suggests that complete quantification is not supported.

Response:

We thank the reviewer for the suggestion. We confirm that the analysis is related to the orientation of filaments with respect to the ventral membrane, where the plasma membrane is in direct contact with the carbon film during cryo-ET data collection. We have replaced "plasma membrane" by "ventral membrane" throughout the manuscript and supplementary information (highlighted in yellow). In addition, we outlined the definition of ventral membrane in **8. Fluorescence image analysis** in the **Methods** section (page 23, line 674-675) as follows: "To determine the angle between the actin meshwork and the ventral membrane (VM, the plasma membrane in direct contact with the culture surface)"; and in **12. Cryo-ET data analysis** in the **Methods** section (page 25, line 750-753) as follows: "The anchored actin filament angles represent the relative orientations between two nearby actin filaments anchored to the ventral membrane (VM, the plasma membrane in direct contact with the carbon film)."

REVIEWER COMMENTS

Reviewer #2 (Remarks to the Author):

Major concerns:

1. The absence of demonstration of glucose-induced insulin release at the 5 min timepoint elicits concern about the cell quality. Detection of insulin release within the first 5 min is routine whilst using healthy cells. The references provided in the response letter do not assess this early timepoint, so the detailed method used by the authors is unknown. The standard expected to support the concept of “biphasic secretion” is perfusion, but many have used a version of this more simplified static culture method with longitudinal sampling (e.g. Srivastava and Goren, *Diabetes*, 2003 52(8): 2049-2056). Did the authors use similar methodology to capture longitudinal insulin release across the timeline associated with biphasic insulin release from beta cells in static culture?

2. The authors state that their secretion results validate that of prior data, and cite Merglen et al. However, Merglen et al document insulin release in units of $\mu\text{g}/10^6$ cells - the authors state release in units of $\text{ng}/10^6$ cells, a substantially reduced insulin release response. The authors also cite Purrello et al for comparable insulin release, however, Purrello et al report their GIS data as % of insulin content, disallowing comparison. The authors also cite comparable release results with that of De Paula et al, but De Paula et al reported insulin release in units of $\text{ng}/10^3$ cells (not 10^6). Taken together, these references suggest that the cells used in the current manuscript may be poorly responsive to stimulatory glucose.

Minor concerns:

Methods describing the rat primary islet beta cell preparation were recently added to the R1 version, but with numerous errors:

- a. How were islets dispersed into single cells? Method states collagenase, which is non-standard. Collagenase is used to digest pancreas to isolate islets, not to dissociate islets. Trypsin, or a similar product, is standard to disperse islet cells. Please correct.
- b. How were single cells sorted to capture only beta cells? Fractionation is a standard method. Please add method used.
- c. Were freshly isolated rat islets used (new text added), or were the frozen islets purchased and somehow thawed, used? If so, why? Please make corrections/deletions to Methods sections 1 and 3. If both types were used, authors must designate the method in each figure legend.

Reviewer #3 (Remarks to the Author):

The authors have addressed my concerns.

Response to reviewers

Reviewer #2 (Remarks to the Author):

Major concerns:

The absence of demonstration of glucose-induced insulin release at the 5 min timepoint elicits concern about the cell quality. Detection of insulin release within the first 5 min is routine whilst using healthy cells. The references provided in the response letter do not assess this early timepoint, so the detailed method used by the authors is unknown. The standard expected to support the concept of “biphasic secretion” is perfusion, but many have used a version of this more simplified static culture method with longitudinal sampling (e.g. Srivastava and Goren, Diabetes, 2003 52(8): 2049-2056). Did the authors use similar methodology to capture longitudinal insulin release across the timeline associated with biphasic insulin release from beta cells in static culture?

Response:

We thank the reviewer for the comments. Given the importance of assessing cell quality throughout the timeline associated with biphasic insulin secretion, we performed two experiments to verify the health and biphasic insulin secretion behavior of both INS-1E β -cells and rat primary β -cells (Supplementary Fig.1 and Supplementary Fig.2).

First, we quantified cumulative insulin secretion over 0-30 minutes under basal conditions (2.8 Glu - 30 min), as well as over 0-5 minutes (16.7 Glu - 5 min) and over 0-30 minutes (16.7 Glu - 30 min) upon glucose stimulation. To further clarify, we have included a more detailed plot of the insulin secretion, indicating 1) timelines of different stimulation conditions, 2) exact insulin amounts at each timepoint and the slopes between adjacent timepoints for better visualization (Please see Figure 1 below). The higher k_2 (red solid line, average secretion rate between 0 min and 5 min under the 16.7 mM glucose) in contrast to k_3 (black dashed line, average secretion rate between 5 min and 30 min under the 16.7 mM glucose) indicates a biphasic pattern in insulin secretion observed in both INS-1E β -cells and rat primary β -cells. We think that this observation may help to further validate the cell quality in our study, and we hope that this implementation of our analyses will meet the reviewer's request.

Figure 1 Insulin secretion under different conditions in INS-1E β -cells (a) and rat primary β -cells (b). The cells transitioned to either basal (2.8 mM glucose) or stimulation solution (16.7 mM glucose) at the zero timepoint. k_1 , k_2 and k_3 denote insulin secretion rates under the following conditions: 2.8 mM glucose over 0-30 minutes (purple solid line), 16.7 mM glucose over 0-5 minutes (red solid line), and 16.7 mM glucose over 5-30 minutes (black dashed line).

Additionally, we conducted Total Internal Reflection Fluorescence (TIRF) measurements every 5 seconds throughout the entire 36-minute duration for both INS-1E β -cells and rat primary β -cells. We examined changes in relative fluorescence intensity of insulin secretory granules in individual β -cells under stimulation with 16.7 mM glucose. The results showed a significant decrease in the rate of fluorescence intensity changes during the 5-30 min stimulation compared to the 0-5 min stimulation in both INS-1E β -cells (from $k_1 = 13.7e-4$ to $k_2 = -4e-4$) and rat primary β -cells (from $k_1 = -20.6e-4$ to $k_2 = -4.5e-4$). This confirms that both types of β -cells exhibit pronounced biphasic insulin secretion characteristics, thus suggesting a fully comparable health state (Please see Figure 2 below).

Figure 2 Quantification of insulin secretory granules in both INS-1E β -cells and rat primary β -cells using live-cell total internal reflection fluorescence (TIRF) during GSIS. k_1 and k_2 denote the rate of fluorescence intensity changes during the 0-5 min and 5-30 min glucose stimulation, respectively.

Following the reviewer's advice to provide more details of the method used, we have now updated the "Insulin secretion level" subsection in the Method section with more detailed information (please, see page 22, line 651-663): "Insulin secretion was quantified as the cumulative insulin release across each treatment period. The INS-1E β -cells and rat primary β -cells were prepared in 12-well plates (Corning) and exposed to KREBS solution containing 2.8 mM glucose for 30 min as the preincubation condition. Subsequently, the cells were incubated: i) for 30 minutes under 2.8 mM glucose, ii) for 5 minutes under 16.7 mM glucose, and iii) for 30 minutes under 16.7 mM glucose conditions, respectively. The supernatant was aspirated at the respective end time point for each

condition. Each sampling well was treated as an independent sample. After the supernatant was removed, cells were washed with 1xPBS once and detached by trypsin. Following centrifuge at 1000 rpm for 3 min, cells were resuspended in 1 ml of complete culture medium and then counted. Insulin levels in the supernatant were measured using an insulin enzyme-linked immunosorbent assay (ELISA) kit (Mercodia). The protein content was subsequently evaluated using Flexstation 3.0 (Molecular Devices).”

2. The authors state that their secretion results validate that of prior data, and cite Merglen et al. However, Merglen et al document insulin release in units of $\mu\text{g}/10\text{e6}$ cells - the authors state release in units of $\text{ng}/10\text{e6}$ cells, a substantially reduced insulin release response. The authors also cite Purrello et al for comparable insulin release, however, Purrello et al report their GISS data as % of insulin content, disallowing comparison. The authors also cite comparable release results with that of De Paula et al, but De Paula et al reported insulin release in units of $\text{ng}/10\text{e3}$ cells (not 10e6). Taken together, these references suggest that the cells used in the current manuscript may be poorly responsive to stimulatory glucose.

Response:

We thank the reviewer for the comments. To clarify, we perform an in-depth comparison of our findings with the results presented in the three papers referenced in our previous response.

Directly comparing the insulin secretion of INS-1E β -cells between our study and Merglen’s (Merglen et al., 2004, *Endocrinology*)¹ is indeed not feasible. Merglen’s study measured the total insulin content in cells ($2.15 \pm 0.11 \mu\text{g}$ insulin/ 10e6 cells), which was not done in our study. However, they detected insulin secretion as a percentage of the total insulin content ($1 \pm 0.2\%$ for 2.5 mM Glu - 30 min and $6.3 \pm 0.6\%$ for 15 mM Glu - 30 min conditions). Thus, the absolute insulin secretion amounts from Merglen’s study can be computed as $21.7 \pm 5.4 \text{ ng}$ and $136.1 \pm 19.8 \text{ ng}/10\text{e6}$ cells under 2.5 mM Glu - 30 min and 15 mM Glu - 30 min conditions, respectively. In our study, insulin secretion of INS-1E β -cells was measured to be 38.1 ± 5.1 and $101.1 \pm 13.6 \text{ ng}$ insulin/ 10e6 cells under 2.8 Glu – 30 min and 16.7 Glu – 30 min conditions, respectively (Please see table 1 below). While there are slight variations in the stimulation conditions between our study and Merglen’s study, the absolute insulin secretion amounts demonstrate comparability, further confirming the quality of our INS-1E β -cells. We also included a table for better comparison as follows:

Table 1 Insulin secretion in INS-1E β -cells measured in our study and Merglen’s (Merglen et al., 2004, *Endocrinology*)¹.

	Subject	Cultivation	Preincubation	Basal	Secretion (ng/ 10e6 cells)	Stimulation	Secretion (ng/ 10e6 cells)
Merglen’s et al ¹	INS-1E β -cells	11.1 mM Glu - 5 days	0 mM Glu - 2 h	2.5 mM Glu - 30 min	21.7 ± 5.4	15 mM Glu - 30 min	136.1 ± 19.8
Ours	INS-1E β -cells	11.1 mM Glu - 48 hours	2.8 Glu - 30 min	2.8 mM Glu - 30 min	38.1 ± 5.1	16.7 mM Glu - 30 min	101.1 ± 13.6

Next, we compare the insulin secretion behavior of isolated rat primary β -cells in our study with isolated rat primary β -cells (Thurmond et al., 2003, *Mol. Endocrinol.*)⁶ and mouse and rat islets (Purrello's study and De Paula's) to validate the health of our rat primary β -cells. In their works, the insulin secretion is presented in terms of fold changes before and after a minimum of 30 min of glucose stimulation, resulting in fairly variable amounts. In fact, insulin secretion is shown to vary greatly, in fold changes, over a range that goes from 1.4 to up to 10 folds, among different cell types, cultivated under different glucose concentrations and upon stimulation with different glucose concentrations (please, see table 2 below). For instance, in Purrello's study (Purrello et al., 1986, *Diabetes*)², insulin secretion in rat islets exhibited 1.4-fold and 9.6-fold changes, both under 22 mM glucose stimulation across varying culture conditions. In De Paula's study (De Paula et al., 2011, *Biol. Res.*)³, rat islets displayed a 8-fold change under 11.1 mM glucose stimulation when cultured with 5.6 mM glucose for 1 h. Overall, it can be observed that lower values in fold changes (i.e., 1.4 to 3.8 folds) are related to a lower delta between the glucose concentrations during stimulation and during cultivation (i.e., 5.3 to 5.6 millimoles/L) (Kolic et al., 2014, *J. Biol. Chem.*, and Wang et al., 2020, *Proc. Natl. Acad. Sci. U. S. A.*; Purrello et al., 1986, *Diabetes* for mouse and rat islets, respectively)^{2,4,5}. Conversely, higher values in fold changes (i.e., 8.0 to 10) are related to a higher delta in glucose concentration (i.e., 8.8 to 16.5 millimoles/L) (Purrello et al., 1986, *Diabetes*; De Paula et al., 2011, *Biol. Res.* for rat islets and Thurmond et al., 2003, *Mol. Endocrinol.* for rat primary β -cells)^{2,3,6}. Noteworthy, the cultivation and stimulation glucose concentration that we used with rat primary β -cells have a delta of 5.6 millimoles/L. Therefore, the value of 1.9 in fold changes that we have obtained, expectedly falls in the same range as the one reported in the literature for analogous experiments.

In addition, in our study the absolute insulin secretion amounts from isolated rat primary β -cells under basal conditions were 2.3 ± 0.3 (ng/10e6 cells, 0.11e6 cells per ml). Similarly, even though related to an unspecified cell number per ml, in Thurmond's study⁸ the absolute insulin secretion amount for the same cell type was reported as 1.3 ± 0.2 ng/ml. (Fig5A, Thurmond's study⁸). It is our opinion that this is indicative of a comparable range, and that therefore such comparisons further validate the biphasic insulin secretion behavior of the rat primary cells in our study, altogether being reminiscent of their healthy state.

Table 2 Insulin secretion in islets or rat primary β -cells measured in different studies.

	Subject	Cultivation	Preincubation	Basal	Secretion	Stimulation	Delta (Stimulation- cultivation)	Fold changes
Ours	Rat primary β -cells	11.1 mM Glu - 24 h	2.8 mM Glu - 30 min	2.8 mM Glu - 30 min	2.3 ± 0.3 (ng/10e6 cells)	16.7 mM Glu - 30 min	5.6 mmol/L	1.9-fold
Purrello et al ²	Rat islets	16.7 mM Glu - 24 h	1.4 mM Glu - 1 h	1.4 mM Glu - 1 h	-	22 mM Glu - 1 h	5.3 mmol/L	1.4-fold
		5.5 mM Glu		1.4 mM Glu	-	22 mM Glu	16.5 mmol/L	9.6-fold

		- 24 h		- 1 h		- 1 h		
De Paula et al ³	Rat islets	5.5 mM Glu - 1 h		5.5 mM Glu - 1 h	-	11.1 mM Glu - 1 h	5.6 mmol/L	8-fold
Wang et al ⁴	Rat islets	11.1 mM Glu - overnight	2.8 mM Glu - 1 h	2.8 mM Glu - 30 min		16.7 mM Glu - 20 min	5.6 mmol/L	2-fold
Kolic et al ⁵	Mouse islets	11.1 mM Glu - 72 h	-	2.8 mM Glu - 30 min	2 ± 0.3 (ng/ml)	16.7 mM Glu - 1 h	5.6 mmol/L	3.8-fold
							5.6 mmol/L	2.5-fold
Thurmond et al ⁶	Rat primary β-cells	11.2 mM Glu - overnight	2 mM Glu - 50 min	2 mM Glu - 1 h	1.3 ± 0.2 (ng/ml)	20 mM Glu - 1 h	8.9 mmol/L	10-fold

Note: Delta denotes the difference in glucose concentration between the stimulation and cultivation conditions. Fold change is computed by dividing the secretion amount under the stimulation condition by that under the basal conditions.

Moreover, Total Internal Reflection Fluorescence (TIRF) microscopy has evolved into a widely applied method for studying biphasic insulin secretion behaviors in pancreatic β-cells. This technique enables the visualization and analysis of individual insulin secretory granule movement and release near the ventral membrane of living pancreatic β-cells (Fig1B and C of Ohara-Imaizumi's study; Fig2b of Qin's study) (Ohara-Imaizumi et al., 2004, *Biochem. J.* and Qin et al., 2017, *EBioMedicine*)^{7,8}. We examined changes in relative fluorescence intensity of insulin secretory granules in individual β-cells under stimulation with 16.7 mM glucose. The results showed a significant decrease in the rate of fluorescence intensity changes during the 5-30 min stimulation compared to the 0-5 min stimulation in both INS-1E β-cells (from $k_1 = 13.7e-4$ to $k_2 = -4e-4$) and rat primary β-cells (from $k_1 = -20.6e-4$ to $k_2 = -4.5e-4$). This confirms that both types of β-cells exhibit pronounced biphasic insulin secretion characteristics, thus suggesting a fully comparable health state (Please see Figure 2 below).

In summary, we think that our experiments collectively affirm the health and biphasic insulin secretion characteristics of the rat primary β-cells utilized in our study. These validations are supported by evidence from insulin secretion (Supplementary Fig. 1), TIRF (Supplementary Fig. 2), immunofluorescence (Supplementary Fig. 5), low-magnification TEM images (Supplementary Fig. 6), and cryo-ET experiments (Fig. 2).

Figure 2 Quantification of insulin secretory granules in both INS-1E β -cells and rat primary β -cells using live-cell total internal reflection fluorescence (TIRF) during GSIS. k_1 and k_2 denote the rate of fluorescence intensity changes during the 0-5 min and 5-30 min glucose stimulation, respectively.

1. Merglen, A. et al. Glucose sensitivity and metabolism-secretion coupling studied during two-year continuous culture in INS-1E insulinoma cells. *Endocrinology* 145, 667–678 (2004).
2. Purrello, F., Vetri, M., Gatta, C., Gullo, D. & Vigneri, R. Effects of high glucose on insulin secretion by isolated rat islets and purified β -cells and possible role of glycosylation. *Diabetes* 38, 1417–1422 (1989).
3. De Paula, F. M. M., Boschero, A. C., Carneiro, E. M., Bosqueiro, J. R. & Rafacho, A. Insulin signaling proteins in pancreatic islets of insulin-resistant rats induced by glucocorticoid. *Biol. Res.* 44, 251–257 (2011).

4. Wang, B. et al. The adaptor protein APPL2 controls glucose-stimulated insulin secretion via F-actin remodeling in pancreatic β -cells. *Proc. Natl. Acad. Sci. U. S. A.* 117, 28307–28315 (2020).
5. Kolic, J., Spigelman, A. F., Smith, A. M., Manning Fox, J. E. & MacDonald, P. E. Insulin secretion induced by glucose-dependent insulintropic polypeptide requires phosphatidylinositol 3-kinase γ in rodent and human β -cells. *J. Biol. Chem.* 289, 32109–32120 (2014).
6. Thurmond, D. C., Gonelle-Gispert, C., Furukawa, M., Halban, P. A. & Pessin, J. E. Glucose-Stimulated Insulin Secretion Is Coupled to the Interaction of Actin with the t-SNARE (Target Membrane SolubleN-Ethylmaleimide-Sensitive Factor Attachment Protein Receptor Protein) Complex. *Mol. Endocrinol.* 17, 732–742 (2003).
7. Ohara-Imaizumi, M. et al. TIRF imaging of docking and fusion of single insulin granule motion in primary rat pancreatic β -cells: different behaviour of granule motion between normal and Goto–Kakizaki diabetic rat β -cells. *Biochem. J.* 381, 13–18 (2004).
8. Qin, T. et al. Munc18b increases insulin granule fusion, restoring deficient insulin secretion in type-2 diabetes human and Goto-Kakizaki rat islets with improvement in glucose homeostasis. *EBioMedicine* 16, 262–274 (2017).

Minor concerns:

Methods describing the rat primary islet beta cell preparation were recently added to the R1 version, but with numerous errors:

a. How were islets dispersed into single cells? Method states collagenase, which is non-standard. Collagenase is used to digest pancreas to isolate islets, not to dissociate islets. Trypsin, or a similar product, is standard to disperse islet cells. Please correct.

Response:

We thank the reviewer for kindly pointing out what were mistakes made by us during the preparation of the manuscript, and for which we sincerely apologize.

Indeed, Collagenase was used to digest pancreas to isolate islets, and 0.1% trypsin was then used to dissociate primary β -cells from the islets. We have now updated the detailed text in Method section (page 20, line 588-608):” To obtain rat primary β -cells, the entire procedure for isolating islets and obtaining single cells was previously described⁹⁻¹³. Briefly, islets were isolated from male Wistar rats (6 weeks old) by collagenase digestion (Worthington) added to 10-fold tissue volume. Digestion was stopped by the addition of ice-cold tissue culture medium (TCM), comprising RPMI-1640 GlutaMAX™-1 medium (Thermo Fisher Scientific) containing 11.1 mM glucose, supplemented with 25 mM HEPES (Thermo Fisher Scientific), 10% heat-inactivated fetal bovine serum (FBS) (Thermo Fisher Scientific), 100 U/ml penicillin and streptomycin (Thermo Fisher Scientific). The digest was filtered through a 500 μ m nylon screen. The additional filtering of undigested residues was repeated twice. Subsequently, isolated islets were washed once with precooled PBS. The islets were added with 6 ml of isotonic Percoll and mixed gently by pipetting, an equal volume of PBS slowly was added to form two density gradients, and centrifuged at 2000 rpm horizontally for 20 min. Utilizing a pipette, the islets were carefully extracted out between two

layers, PBS was added and the resuspension centrifuged at 1000 rpm for 3 min. After discarding the supernatant, islets were resuspended in 0.1% trypsin (Thermo Fisher Scientific) and digested for 5-10 min. After trypsinization stopped by TCM, the cell suspension was centrifuged at 1000 rpm for 3 min and resuspended in 2.8 mM glucose. The cell suspension was then submitted to auto-fluorescence-activated cell sorting (FACS) with 100mW at 488 nm excitation. The β -cells were collected at 510-550 nm fluorescence emission. The sorted β -cells were cultured on dish (Falcon) at 37°C in 5% CO₂ for further experiments.”

b. How were single cells sorted to capture only beta cells? Fractionation is a standard method. Please add method used.

Response:

We thank the reviewer for the suggestions. Fractionation and FACS sorting were used to obtain the β -cells. We updated the detailed text in Method section (page 20, line 588-608):” To obtain rat primary β -cells, the entire procedure for isolating islets and obtaining single cells was previously described⁹⁻¹³. Briefly, islets were isolated from male Wistar rats (6 weeks old) by collagenase digestion (Worthington) added to 10-fold tissue volume. Digestion was stopped by the addition of ice-cold tissue culture medium (TCM), comprising RPMI-1640 GlutaMAX™-1 medium (Thermo Fisher Scientific) containing 11.1 mM glucose, supplemented with 25 mM HEPES (Thermo Fisher Scientific), 10% heat-inactivated fetal bovine serum (FBS) (Thermo Fisher Scientific), 100 U/ml penicillin and streptomycin (Thermo Fisher Scientific). The digest was filtered through a 500 μ m nylon screen. The additional filtering of undigested residues was repeated twice. Subsequently, isolated islets were washed once with precooled PBS. The islets were added with 6 ml of isotonic Percoll and mixed gently by pipetting, an equal volume of PBS slowly was added to form two density gradients, and centrifuged at 2000 rpm horizontally for 20 min. Utilizing a pipette, the islets were carefully extracted out between two layers, PBS was added and the resuspension centrifuged at 1000 rpm for 3 min. After discarding the supernatant, islets were resuspended in 0.1% trypsin (Thermo Fisher Scientific) and digested for 5-10 min. After trypsinization stopped by TCM, the cell suspension was centrifuged at 1000 rpm for 3 min and resuspended in 2.8 mM glucose. The cell suspension was then submitted to auto-fluorescence-activated cell sorting (FACS) with 100mW at 488 nm excitation. The β -cells were collected at 510-550 nm fluorescence emission. The sorted β -cells were cultured on dish (Falcon) at 37°C in 5% CO₂ for further experiments.”

9. Scarl, R. T., Koch, W. J., Corbin, K. L. & Nunemaker, C. S. Isolation and assessment of pancreatic islets versus dispersed beta cells: A straightforward approach to examine cell-cell communication. *Methods Mol. Biol.* 2346, 151–164 (2021).

10. Rivera, J. F., Costes, S., Gurlo, T., Glabe, C. G. & Butler, P. C. Autophagy defends pancreatic β cells from human islet amyloid polypeptide-induced toxicity. *J. Clin. Invest.* 124, 3489–3500 (2014).

11. Wang, Z. et al. Live-cell imaging of glucose-induced metabolic coupling of β and α cell metabolism in health and type 2 diabetes. *Commun. Biol.* 4, 594 (2021).

12. Stangé, G., Van De Casteele, M. & Heimberg, H. Purification of rat pancreatic β -cells by fluorescence-activated cell sorting. *Diabetes Mellitus* 015–022 (Humana Press, 2003).

13. Van De Winkel, M., Maes, E. & Pipeleers, D. Islet cell analysis and purification by light scatter and autofluorescence. *Biochem. Biophys. Res. Commun.* 107, 525–532 (1982).

c. Were freshly isolated rat islets used (new text added), or were the frozen islets purchased and somehow thawed, used? If so, why? Please make corrections/deletions to Methods sections 1 and 3. If both types were used, authors must designate the method in each figure legend.

Response:

We thank the reviewer for raising this important question. Actually, we have purchased rat primary β -cells from Shanghai Zhong Qiao Xin Zhou Biotechnology Co., Ltd. These cells were dissociated from freshly isolated islets from Wistar rats and immediately used for subsequent experiments.

REVIEWERS' COMMENTS

Reviewer #2 (Remarks to the Author):

Though most concerns were adequately addressed, one was not and remains a fundamental issue to be clarified in the main text:

In the prior round of review, the authors were asked to address the following:

"c. Were freshly isolated rat islets used (new text added), or were the frozen islets purchased and somehow thawed, used? If so, why? Please make corrections/deletions to Methods sections 1 and 3. If both types were used, authors must designate the method in each figure legend."

The authors responded by stating "Actually, we have purchased rat primary β -cells from Shanghai Zhong Qiao Xin Zhou Biotechnology Co., Ltd. These cells were dissociated from freshly isolated islets from Wistar rats and immediately used for subsequent experiments."

This is confusing--how did the authors receive from the company freshly isolated and dissociated beta cells, and not frozen stocks? To ensure reproducibility by others in future, please add clarifying text:

State at the beginning of their new Methods section on page 20, prior to lines 588-608 that at the Shanghai company, rat islets were isolated and beta cells dissociated and captured by FACS, and those (fresh? or frozen?) sorted beta cells were purchased (provide vendor, catalog #, etc.) by the authors labs for subsequent thawing/culture and study.

This is important so that researchers buy the right cells from Shanghai, or that they isolate their own cells in their labs, when trying to springboard from the findings presented in this paper.

This clarification is critical to meet the standards for reproducibility, for the scientific field to be able to reproduce the findings and move the work forward.

Response to reviewers

Reviewer #2 (Remarks to the Author):

Though most concerns were adequately addressed, one was not and remains a fundamental issue to be clarified in the main text:

In the prior round of review, the authors were asked to address the following:

"c. Were freshly isolated rat islets used (new text added), or were the frozen islets purchased and somehow thawed, used? If so, why? Please make corrections/deletions to Methods sections 1 and 3. If both types were used, authors must designate the method in each figure legend."

The authors responded by stating "Actually, we have purchased rat primary β -cells from Shanghai Zhong Qiao Xin Zhou Biotechnology Co., Ltd. These cells were dissociated from freshly isolated islets from Wistar rats and immediately used for subsequent experiments."

This is confusing--how did the authors receive from the company freshly isolated and dissociated beta cells, and not frozen stocks? To ensure reproducibility by others in future, please add clarifying text:

State at the beginning of their new Methods section on page 20, prior to lines 588-608 that at the Shanghai company, rat islets were isolated and beta cells dissociated and captured by FACS, and those (fresh? or frozen?) sorted beta cells were purchased (provide vendor, catalog #, etc.) by the authors labs for subsequent thawing/culture and study.

This is important so that researchers buy the right cells from Shanghai, or that they isolate their own cells in their labs, when trying to springboard from the findings presented in this paper.

This clarification is critical to meet the standards for reproducibility, for the scientific field to be able to reproduce the findings and move the work forward.

Response:

We thank the reviewer for the suggestions. We purchased and received freshly isolated and dissociated rat primary β -cells from Shanghai Zhong Qiao Xin Zhou Biotechnology Co., Ltd., located near our lab. These cells were cultured in T25 flasks with tissue culture medium and delivered to our facility, where they were placed in a 37°C incubator upon arrival.

We agree that clarification on the source of fresh rat primary β -cells is important for ensuring the reproducibility of our experimental results. We have updated the details in the Methods section as follows (page 20, lines XX-XXX): "At the company (Shanghai Zhong Qiao Xin Zhou Biotechnology

Co., Ltd.), rat islets were isolated and β -cells were dissociated and then captured by Fluorescence-Activated Cell Sorting (FACS). Those freshly sorted β -cells were purchased (PRI-RAT-00136) for subsequent culture and study."